# SOA: Strategic Operator Adaptation for Accelerating Joint In-Context Prompt Optimization

## Abstract

Strategic prompt-tuning in Large Language Models (LLMs) presents a formidable challenge that requires substantial resources and expert human input. Prior research has treated the tuning of prompt instructions and few-shot examples as distinct and separate problems, resulting in sub-optimal performance. This work overcomes this limitation by introducing a joint prompt-tuning approach that optimizes both the instruction and examples simultaneously. However, formulating such an optimization in the discrete and high-dimensional space of natural language poses significant challenges in terms of convergence and computational efficiency. To address these challenges, we propose, `SOA`, a novel Strategic Operator Adaptation framework, designed to accelerate the optimization process by strategically employing a variety of operators to traverse the prompt space effectively for both zero-shot and few-shot scenarios. `SOA` features a quad-phased design that fully exploits the potential of each phase, alternating between global traversal and local optimization to strike a balance between exploration and exploitation in this complex space. By adaptively selecting the best operators for traversal and actively pruning less desirable candidates, `SOA` is able to identify the best combination of instructions and examples while minimizing inference costs. We have conducted a comprehensive evaluation across 35 benchmark tasks, and the results show that `SOA` significantly outperforms state-of-the-art baseline methods by a large margin, achieving an average task performance improvement of **35.47**% while significantly reducing computational costs by **58.67**% in the BIG-Bench-Hard tasks. [1]

## 1 Introduction

Large Language Models (LLMs) have exhibited extraordinary performance across various domains and tasks (Bubeck et al., 2023; Yang et al., 2023b), largely owing to their remarkable ability of in-context learning (ICL). Prompt engineering seeks to craft effective prompts that unleash the complete capabilities of LLMs. It is becoming an increasingly popular option for quickly adapting LLMs for downstream tasks due to its compatibility with black-box APIs (e.g., GPT-4 (OpenAI, 2023) and PaLM 2 (Chowdhery et al., 2022)), and its cost-effectiveness compared to the conventional fine-tuning paradigm. The two most typical prompting strategies are *zero-shot prompting* which contains a task instruction and a query question, and *few-shot prompting* which includes additional illustrative examples. A good prompt design can substantially improve LLM's performance (Zhu et al., 2023); however, manual prompt tuning and selection is a heavily *time-consuming* process that requires significant human effort and expert knowledge.

Automating prompt tuning is a non-trivial optimization task that involves discrete variables and complex high-dimensional spaces (Zhou et al., 2023). Existing studies treat the tuning of prompt instruction and in-context examples as separate tasks: one line of research (Pryzant et al., 2023; Chen et al., 2023; Yang et al., 2023a; Guo et al., 2023) takes the zero-shot prompting approach (Kojima et al., 2022) to focus on *tuning a short instruction* that comprises one or few sentences; while the other line of work (Liu et al., 2021; Lu et al., 2021; 2022; Zhang et al., 2022b; An et al., 2023) emphasizes more the importance of few-shot examples (Brown et al., 2020) and seeks to *selecting the best set of examples* from a pre-defined dataset given a *fixed*

---

[1]The source code and datasets are ready to be publicly available for research purposes.

instruction. Although such treatment reduces the problem complexity, it overlooks the significance of the interplay between instruction and in-context examples, resulting in *sub-optimal* performance (Hsieh et al., 2023).

In this work, we tackle two important challenges of past prompt-tuning strategies: (i) how to design an automatic pipeline that effectively traverses the high-dimensional joint space of instructions and examples, steering clear of local minima and ensuring continuous performance enhancement? (ii) what strategies can be employed to accelerate joint prompt tuning, enabling fast convergence with a reasonable level of computational cost?

We first introduce a joint prompt-tuning problem that simultaneously optimizes the prompt instruction and examples as a whole. As illustrated in Figure 1, our formulation does not impose any restrictions or assumptions on the style (zero-shot or few-shot) of the prompt, thereby unlocking the full potential of prompt traversal in contrast to previous instruction-only optimization methods (Zhou et al., 2023; Pryzant et al., 2023; Chen et al., 2023; Guo et al., 2023; Fernando et al., 2023), Notably,

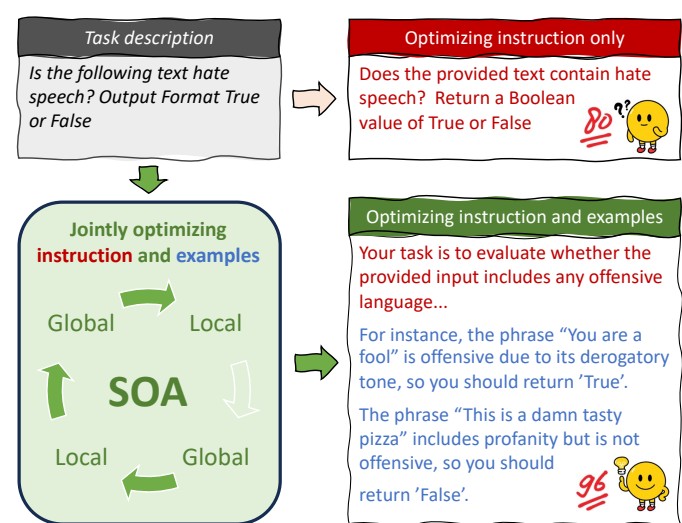

Figure 1: An illustrative example of the joint prompt-tuning of instruction and examples, which shows better performance than instruction-only optimization.

our formulation not only enables innovative instruction exploration but is also capable of producing novel examples to enhance the generalizability of LLMs. Consequently, our optimal prompt is highly adaptive and flexible to any style from a simple zero-shot instruction-only prompt to an elaborative few-shot prompt with detailed examples, depending on the specific task at hand.

We then propose a novel Strategic Operator Adaptation (`SOA`) framework aimed at accelerating joint prompt optimization in high-dimensional spaces while minimizing inference costs. `SOA` introduces a quad-phased design that fully excavates all potentials of each phase and alternates between global traversal and local optimization, striking an optimal balance between exploration and exploitation within the challenging high-dimensional space. This is accomplished by thoroughly analyzing a suite of LLM operators to pinpoint their unique strengths and features. By adaptively choosing the best operators for traversal and actively pruning undesired candidates, `SOA` can achieve optimal performance while accelerating convergence speed. Additionally, we integrate two innovative designs to enhance the performance and efficiency of `SOA`. Firstly, we introduce a task-aware similarity metric based on performance-based vectors and hamming distance, proving more effective than traditional lexical similarity metrics. Secondly, we implement adaptive phase stop criteria that ensure maximum performance improvement with the current operator before transitioning to the next, optimizing the overall framework efficiency.

We conduct an extensive evaluation on a total number of 35 benchmark tasks and empirically show that `SOA` demonstrates substantial improvements compared to state-of-the-art (SOTA) methods, including *APE* (Zhou et al., 2023), *APO* (Pryzant et al., 2023), *OPRO* (Yang et al., 2023a), *PromptBreeder* (Fernando et al., 2023), *EvoPrompt* (Guo et al., 2023), and *AELP* (Hsieh et al., 2023), and these advancements are achieved with the lowest computational cost among all baselines. For harder tasks like BBH, `SOA` introduces an average of **35.47%** task accuracy improvement while reducing **58.67%** of inference costs compared to SOTA methods.

## 2 Problem Description

Considering the task $\mathcal{T}$ specified by a dataset $\mathcal{D} = (\mathcal{Q}, \mathcal{A})$ of input/output pairs, the LLM $\mathcal{L}$ produces the corresponding output $\mathcal{A}$ via prompting with the concatenation of prompt $\mathcal{P}$ and a given input $\mathcal{Q}$, i.e., $[\mathcal{P}; \mathcal{Q}]$.

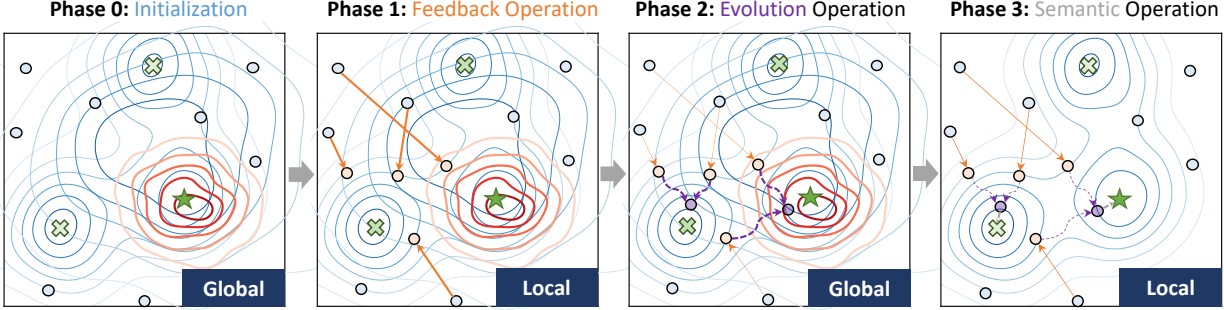

**Phase 0:** Initialization     **Phase 1:** Feedback Operation     **Phase 2:** Evolution Operation     **Phase 3:** Semantic Operation

★ Global Optima    ⊗ Local Optima    ○ In-Context Prompt

Figure 2: `SOA` framework aims at surfacing the globally optimal in-context prompt (instruction and example combination) by iteratively traversing the high-dimensional discrete space from a pool of candidates and pruning unpromising candidates along the process. `SOA` realizes strategic traversal by introducing a dual exploration-exploitation strategy, i.e., "global exploration" → "local exploitation" → "global exploration" → "local exploitation" where operators are applied strategically during phases and transitions between phases are determined adaptively at run time. `SOA` achieves strategic pruning by employing a greedy-based strategy looking at the candidate's performance.

The objective of prompt optimization is to design the best natural language prompt $\mathcal{P}^*$ that maximizes the performance of $\mathcal{L}$ on $\mathcal{T}$.

Typically, an ideal prompt $\mathcal{P}$ consists of *instruction*, denoted by $\mathcal{I}$ and *examples* denoted by $\mathcal{E}$ as in-context learning (ICL) demonstrations. Our goal of joint prompt optimization is to search for the optimal prompt $\mathcal{P}^*_{(\mathcal{I},\mathcal{E})}$ given $\mathcal{L}$ that maximizes the performance towards a performance metric function $\mathcal{F}$ (e.g., accuracy). This can be formally defined as the following optimization problem:

$$\mathcal{P}^*_{(\mathcal{I},\mathcal{E})} = \underset{\mathcal{P}_{(\mathcal{I},\mathcal{E})} \in \mathcal{X}}{\arg\max} \, \mathbb{E}_{(\mathcal{Q},\mathcal{A})} \left[ \mathcal{F}(\mathcal{P}_{(\mathcal{I},\mathcal{E})}; \mathcal{Q}, \mathcal{A}) \mid \mathcal{L} \right], \tag{1}$$

where $\mathcal{X}$ denotes the sample space for a natural language prompt, a discrete and intractable space of arbitrarily large dimension, which makes the optimization problem in Eq. 1 extremely difficult.

## 3 Proposed Methodology: Strategic Operator Adaptation (`SOA`)

We propose a novel framework, `SOA`, that leverages a variety of operators to traverse the prompt space for both zero-shot and few-shot to surface the best instruction and examples combination. By adaptively choosing the best operators for traversal and actively pruning undesired candidates, `SOA` can achieve optimal performance while accelerating convergence speed.

`SOA` adaptively switches between two distinct traversal strategies to balance speed and performance: (1) *Exploration*, where a specific subset of operators are leveraged for a *global* search to broadly explore the entire solution space and prevent entrapment in locally optimal solutions; (2) *Exploitation*, which involves the use of another set of operators for local search to expedite convergence and improve efficiency. Instead of blindly using a fixed set of operators following a pre-defined sequence, or randomly selecting operators, `SOA` aims to organize multiple operators adaptively and strategically. The application of the optimal operator at the right time, combined with active greedy-based pruning, ultimately leads to accelerated performance in both task accuracy and convergence speed.

### 3.1 Operator Elaboration

Following the insight of leveraging global search and local search, we introduce five operators that can be categorized as global operators and local operators. The three *global* operators are:

- **Lamarckian Operator** is a reverse-engineering operator $\mathcal{O}_L$ that passes input-output pairs to an LLM agent and asks the agent to "reverse-engineer" the instruction.

- **Estimation of Distribution Operator (EDA)** is a function operator $\mathcal{O}_E$ that takes in a group of candidates and inquires an LLM agent to output a new candidate by studying the input group. If the input group is chosen by prioritizing distinctiveness, we call it EDA + Index (EDA+I).

- **Crossover Operator (CR)** is a function operator $\mathcal{O}_C$ that takes two parents and asks an LLM agent to generate a new candidate mixing the traits of both parents. If the parents are chosen by prioritizing distinctiveness between them, we call it Crossover + Distinct (CR + D).

The two *local* operators are:

- **Feedback Operator** is a function operator $\mathcal{O}_F$ that utilizes two LLM agents. $\mathcal{O}_F$ first passes in input-output pairs of tasks where an existing candidate makes mistakes to an "Examiner" agent, whose task is to examine the places of mistakes and provide remediation strategies. It then uses an "Improver" agent that takes the remediation strategies and applies them to the existing candidate to generate a new candidate.

- **Semantic Operator** is a function operator $\mathcal{O}_S$ that takes in an existing candidate, and modifies the candidate lexically while preserving its semantic meaning.

To better harness the power of these operators, we compare them along the following five features that are critical to our exploration-exploitation strategy in terms of performance and efficiency:

- **Add or remove examples?** This examines whether an operator can add or remove few-shot examples, to traverse the entire prompt space of both zero-shot and few-shot.

- **Probability of improvement.** This evaluates the probability (successful rate) of an operator that brings performance improvement (gain) after iterations. The higher, the better.

- **Convergence speed.** This metric evaluates how fast (in terms of iterations) an operator needs to traverse to the current candidate's local minimum solution.

- **Two or more parents?** This indicates whether an operator needs two or more parents, which has the potential to combine traits from diverse ancestor lines, enhancing global exploration capability.

- **Inference cost per operation.** This is the number of inference calls needed to perform a specific operator acted by an LLM agent.

Table 1: Elaborated feature analysis of operators. The number of black dots (●) represents a relative comparison among operators in terms of performance gain, convergence, and computational cost.

| Operator | Add examples | Remove examples | Two or more parents | Prob of improvement | Convergence speed | Inference cost |
|---|---|---|---|---|---|---|
| Lamarckian | ✓ | - | - | - | - | ● |
| Feedback | ✓ | ✓ | - | ●● | ●●●● | ●● |
| EDA (EDA+I) | - | - | ✓ | ●●●● | ●● | ● |
| Crossover (CR+D) | - | - | ✓ | ●●●● | ●● | ● |
| Semantic | - | ✓ | - | ●●● | ●●● | ● |

We conducted a series of experiments (ran each operator 100 times based on 4 different initialization settings) to assess the performance of each operator regarding the five features, aiming at obtaining a comprehensive understanding of the inherent strengths and weaknesses of each operator. This allows us to select effective operators to find optimal solutions in an accelerated manner. As shown in Table 1, we observe that the Lamarckian operator is a crucial operator that introduces diverse samples, enabling the addition of examples

for global initialization. The feedback operator leads to faster convergence (four ●) for exploitation and facilitates the addition or removal of examples but it requires two API/inference calls (two ●), higher than the other operators (one ●). EDA and Crossover operators share similar characteristics that indicate a higher probability of improvement (four ●) than the semantic operator (three ●) and feedback operator (two ●) in exploring the global space. For a more in-depth discussion on operators, please refer to Appendix C.1 and C.2.

### 3.2 `SOA` Framework

#### 3.2.1 Phase 0: Global Initialization

Our objective is to create diverse candidates as the initial pool to explore the vast joint space of instruction and example. We provide two types of initialization based on the availability of data (*input/output pair*) and human expert knowledge (*prompt example*).

- **Reverse Engineer from input/output pairs.** Given a set of input/output pairs $S = \{(Q_1, A_1), ..., (Q_m, A_m)\}$ from the training set $\mathcal{D}_{\text{train}}$ for the task $\mathcal{T}$, we define an LLM agent to apply Lamarckian Operator $\mathcal{O}_L$ to *reverse engineer* the prompt from provided demonstrating pairs.
- **Human expert prompt example.** This way allows humans to jump-start the tuning process by incorporating prior knowledge. We also perform the semantic operator $\mathcal{O}_S$ to enhance the diversity of the initial pool.

#### 3.2.2 Phase 1: Local Feedback Operation

While an initial phase (Phase 0) may result in a diverse pool, each candidate could still be distant from its local optimal solution. We want to arrive at the local optimal to exhaust the potential of the candidates. To address this, we employ the Feedback Operator $\mathcal{O}_F$ to expedite each candidate's convergence towards their local minimums, leveraging the "gradient" information. This involves the introduction of an LLM *Examiner*, which scrutinizes instances where the current candidate falls short, and subsequently offers improvement guidance. Such information is taken as the feedback gradient and is further utilized by an LLM *Improver*, to generate new candidates by local exploitation. These new candidates contain global information inherited from the previous phase and can thus be regarded as better initialization for the next optimization phase.

#### 3.2.3 Phase 2: Global Evolution Operation

Phase 1 provides a more refined set of candidates, while some of them might be stuck in local optima. To address this issue, we prioritize exploration rather than exploitation in Phase 2, which helps to escape from these restricted localities by conducting a global search. We leverage LLM agents that are inspired by genetic evolution, specifically EDA (EDA-I) operators $\mathcal{O}_E$ and CR (CR-D) operators $\mathcal{O}_C$ to facilitate the increased interaction of genetic information among candidates on a larger global scale. Rather than employing cosine similarity as distance metrics, we adopt the Hamming distance (see more discussions in Section 3.3) for calculating similarity on performance-based vectors such that Phase 2 can promote greater diversity during iteration.

#### 3.2.4 Phase 3: Local Semantic Operation

Upon completing Phase 2's exploration, Phase 3 employs local exploitation to hasten the "last mile" of convergence. As the concluding phase of `SOA`, the performance of the pool is notably optimized at this stage relative to earlier phases. Consequently, the Semantic operator $\mathcal{O}_S$ is selected to expedite more cost-effective exploitation of the candidates. Finally, we identify the best candidate as our ultimate optimal prompt and assess its performance on the testing dataset $\mathcal{D}_{\text{test}}$. The workflow of `SOA` framework is shown in Algorithm 1.

### 3.3 `SOA` Design Schemes

Within the `SOA` framework, we propose two novel design schemes to improve performance and efficiency.

---

**Algorithm 1** `SOA` for Accelerating Joint In-Context Prompt Optimization

---

1: **requirements**: size of pool $n$, a dev set $\mathcal{D}_{\text{dev}}$, score function $\mathcal{F}$ on the base LLM $\mathcal{L}$, phase improvement $t$ and threshold $t^*$ and minimum run time for phases $\mathcal{K}_i$, designed operators $\mathcal{O}_L$, $\mathcal{O}_F$, $\mathcal{O}_E$, $\mathcal{O}_C$ and $\mathcal{O}_S$

2: **initialization**: generate diverse initial prompts $\mathcal{P}^0 = \{p_1^0, ..., p_n^0\}$ by $\mathcal{O}_l$ with input/output pairs or $\mathcal{O}_s$ with existing prompt, and evaluate initial scores $\mathcal{S}^0 \leftarrow \{s_i^0 = \mathcal{F}(p_i^0, \mathcal{D}_{\text{dev}})\}$      // Phase 0

3: **while** $t < t^*$ or $k \leq \mathcal{K}_1$ **do**      // Phase 1

4:      ***Local Feedback Operation:*** generate new prompts by feedback operator, $\mathcal{P}_t \leftarrow \mathcal{O}_f(\mathcal{P}^0)$, evaluate $\mathcal{S}_t \leftarrow \mathcal{F}(\mathcal{P}^0, \mathcal{D}_{\text{dev}}))$, and update the pool $\mathcal{P}^1 \leftarrow \{\mathcal{P}_t, \mathcal{P}^0\}$, and score set $\mathcal{S}^1 \leftarrow \{\mathcal{S}_t, \mathcal{S}^0\}$

5: **while** $t < t^*$ or $k \leq \mathcal{K}_2$ **do**      // Phase 2

6:      ***Global Evolution Operation:*** select input prompts from the current pool, $\{p_{r_1}, ..., p_{r_k}\} \in \mathcal{P}^1$, generate a new prompt by performing EDA operators $p_t \leftarrow \mathcal{O}_e(p_{r_1}, ..., p_{r_k})$ or crossover operators $p_t \leftarrow \mathcal{O}_c(p_{r_1}, ..., p_{r_k})$, evaluate on $\mathcal{D}_{\text{dev}}$, $s_t \leftarrow \mathcal{F}(p_t, \mathcal{D}_{\text{dev}})$, and update $\mathcal{P}^2 \leftarrow \{\mathcal{P}^1, p_t\}$ and $\mathcal{S}^2 \leftarrow \{\mathcal{S}^1, s_t\}$

7: **while** $t < t^*$ or $k \leq \mathcal{K}_3$ **do**      // Phase 3

8:      ***Local Semantic Operation:*** generate new prompts by the semantic operator $\mathcal{P}_t^* \leftarrow \mathcal{O}_s(\mathcal{P}^2)$, evaluate $\mathcal{S}_t^* \leftarrow \mathcal{F}(\mathcal{P}^2, \mathcal{D}_{\text{dev}}))$, and update $\mathcal{P}^3 \leftarrow \{\mathcal{P}_t^*, \mathcal{P}^2\}$, and $\mathcal{S}^3 \leftarrow \{\mathcal{S}_t^*, \mathcal{S}^2\}$

9: **return** the optimal prompt $p^*$, from the final pool $\mathcal{P}^3$: $p^* \leftarrow \arg\max_{p \in \mathcal{P}^3} \mathcal{F}(p, \mathcal{D}_{\text{dev}})$

---

*Design 1: Performance vector with Hamming distance.* Operators like EDA and Crossover function optimally when parents exhibit distinct attributes. In terms of evaluating similarity scores, we adhere to the principle that similarity should be gauged based on the performance of the prompts rather than their linguistic or semantic similarities. Inspired by this intuition, we propose to construct candidate vectors based on individual performance on the evaluation dataset, named "performance vectors". To exemplify, in an evaluation dataset comprising five elements, a candidate answering the first three queries correctly and the final two incorrectly would feature a vector representation of $[1, 1, 1, 0, 0]$.

Rather than calculating the cosine similarity of embedding space, we propose to compute candidate similarity scores by *Hamming distance*, which calculates the distance between two vectors of equal length by examining the number of positions at which the corresponding symbols are different. This way ensures that one candidate is more likely to be paired with a candidate that does not contain the same mistakes, and thereby generates a diverse pool with a more diverse set of genetic information.

*Design 2: Adaptive Phase Stop Criteria.* To ensure that each operation phase is fully conducted before we transition to the next, the decision to move to the next phase is made adaptively based on two primary criteria.

- **Performance Gain.** If no performance gain manifests after implementing the operators in a particular phase, it's indicative that the candidate has been thoroughly optimized by the operators. Consequently, we transition to the next phase.

- **Operator-specific Tolerance.** As operators inherently vary, for more localized operators, e.g., the feedback operator, which has high improvement probabilities, if no performance gain is perceived, it indicates applying the operator for another iteration will unlikely bring additional benefits. However, global operators might have low initial improvement probabilities but are capable of accessing broader branches worth exploring. Therefore, we should be more patient by assigning greater *tolerance*, which will run them at least for a pre-defined time even without immediate performance improvement. More details about the stop criteria can be found in Appendix C.2.

## 4 Experiments

### 4.1 Experimental Setup

**Tasks and Datasets.** We curate 35 benchmark tasks from three domains for thorough experiments: 8 Big Bench Hard (**BBH**) (Suzgun et al., 2022a); 3 NLP detection tasks, including **Ethos** (Mollas et al., 2021), **Liar** (Wang, 2017), and **Sarcasm** (Farha & Magdy, 2020); 24 instruction induction tasks (Honovich et al., 2022). The task and dataset details are in Appendix E.

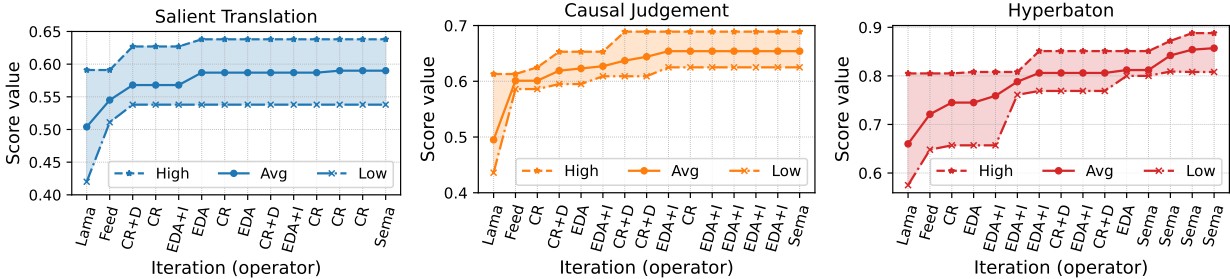

Figure 3: Iteration history of score values with different operators during optimization. The local and global operators are strategically and adaptively selected by each iteration.

**Baselines.** We evaluate `SOA` against a variety of LLM-based approaches that have achieved state-of-the-art performance in prompt optimization:

- **APE** (Zhou et al., 2023) and **APO** (Pryzant et al., 2023): APE utilizes an iterative Monte Carlo Search strategy that emphasizes *exploration*, while APO emphasizes *exploitation*, which harnesses incorrect instances as feedback gradient to refine the original prompt.

- **OPRO** (Yang et al., 2023a): OPRO leverages LLM as optimizers to generate better instruction via meta-prompt, solution-score pairs, and task descriptions.

- **PromptBreeder** (Fernando et al., 2023), **EvoPrompt** (Guo et al., 2023) and **AELP** (Hsieh et al., 2023): these methods connect LLMs with evolution algorithms (EAs) to tackle prompt optimization tasks. Specifically, EvoPrompt implements EAs using genetic algorithm (Holland, 1992) and differential evolution (Storn & Price, 1997), while PromptBreeder introduces multiple operators inspired by thinking styles. AELP focuses on long prompt optimization by mutating on a sentence level with a history-guided search.

**Implementation Details.** We utilized GPT-3.5 to develop LLM agents capable of performing various operators. We set up training, development, and testing datasets, select the prompt with the highest score on the dev set, and report its score on the testing set. We compared the performance of multiple LLM agent models, including PaLM 2, Claude 2, Llama2/3, and Mistral models. More details are provided in Appendix E.

### 4.2 Main Results

**BBH Tasks.** Following the practice of AELP (Hsieh et al., 2023), we conduct 8 BBH tasks to evaluate the performance of `SOA` holistically. We consider two initialization schemes `SOA`-pair and `SOA`-example and report the final best prompt results in Table 2. `SOA` demonstrates substantial improvements compared to state-of-the-art methods, achieving an average improvement of over AELP (**60.4**%↑), EvoPromopt (**21.7**%↑), and OPRO (**24.3**%↑). In terms of computational cost, `SOA` consumes **40%** of AELP, **80%** of EvoPrompt, **4%** of OPRO. Fig. 3 depicts the iterative history of prompt evolution, emphasizing the score variations for the best candidate, worst candidate, and the pool's average across iterations. It has been observed that the Feedback operator yields a performance boost within a single iteration and rarely introduces continual improvements. Global operators such as EDA and Crossover aid in escaping local minima and offering additional performance leaps (refer to Hyperbaton). This observation aligns with our initial operator analysis. The success of `SOA` lies in the organic organization of these operators, effectively harnessing their advantages to optimize performance and accelerate convergence.

**Detection Tasks.** To present a more extensive comparison, we adopted the configuration outlined in APO (Pryzant et al., 2023) and conducted a comparative analysis against it across three tasks. It should be noted that data for the fourth task mentioned in the original paper is unavailable. As shown in Table 3, `SOA` exhibits marginally superior performance to APO in relatively simple tasks such as Ethos (by 1%) and Sarcasm (by 4.7%). However, for more complex tasks such as Liar, `SOA` demonstrates a significant improvement of

Table 2: Testing performance of the optimal prompt on 8 representative tasks from BBH.

| Method | Causal Judgement | Dis -ambiguation | Dyck Languages | Formal Fallacies | Hyperbaton | Logical Five | Color Reasoning | Salient Translation |
|---|---|---|---|---|---|---|---|---|
| OPRO | 71.94 | 71.53 | 36.73 | 49.51 | 75.92 | 50.00 | 65.55 | 43.88 |
| EvoPrompt | 67.24 | 53.70 | **47.96** | 50.81 | 74.79 | 61.40 | 60.90 | 47.58 |
| AELP | 77.77 | 64.79 | 10.67 | 58.25 | 53.74 | 73.49 | 68.14 | 41.43 |
| SOA-pair | 72.13 | **72.37** | 8.060 | **58.87** | 86.02 | 48.19 | 60.52 | **49.19** |
| SOA-example | **89.09** | 68.47 | 46.77 | 58.65 | **87.51** | **86.29** | **80.64** | 47.59 |
| Over AELP | 14.5% ↑ | 11.7% ↑ | 338.3% ↑ | 1.1% ↑ | 62.8% ↑ | 17.4% ↑ | 18.3% ↑ | 18.7% ↑ |
| Over EvoPrompt | 32.5% ↑ | 34.8% ↑ | -2.5% | 15.9% ↑ | 17.0% ↑ | 40.5% ↑ | 32.4% ↑ | 3.4% ↑ |
| Over OPRO | 23.8% ↑ | 1.2% ↑ | 27.3% ↑ | 18.9% ↑ | 15.3% ↑ | 72.6% ↑ | 23.0% ↑ | 12.1% ↑ |

27.5% compared to APO. Moreover, we have also provided results for SOA using GPT-4, which demonstrated performance comparable to those of SOA employing GPT-3.5.

**Instruction Induction Tasks.** To compare SOA-generated prompts with manually added few-shot examples, we evaluated the optimized prompt from SOA against the best prompts from APE-fewshot (Zhou et al., 2023) and PromptBreeder-fewshot (Fernando et al., 2023) on APE's 24 instruction induction tasks. The results show that SOA outperforms APE in 17 out of 24 tasks and PromptBreeder in 18 out of 24 tasks. The Appendix F.1 provides complete experimental results. Fig. 4 shows that

Table 3: Testing performance on three detect tasks used by APO.

| Method | Ethos | Liar | Sarcasm |
|---|---|---|---|
| APO | 0.95 | 0.51 | 0.85 |
| SOA (GPT-3.5) | 0.96 | 0.65 | 0.87 |
| SOA (GPT-4) | 0.96 | 0.69 | 0.89 |

few-shot methods do not always outperform zero-shot methods, highlighting the need for a joint in-context prompt search. Moreover, we observed that the prompts generated by SOA are easier to interpret and align better with the task description. Appendix F.2 provides more detail on prompt quality.

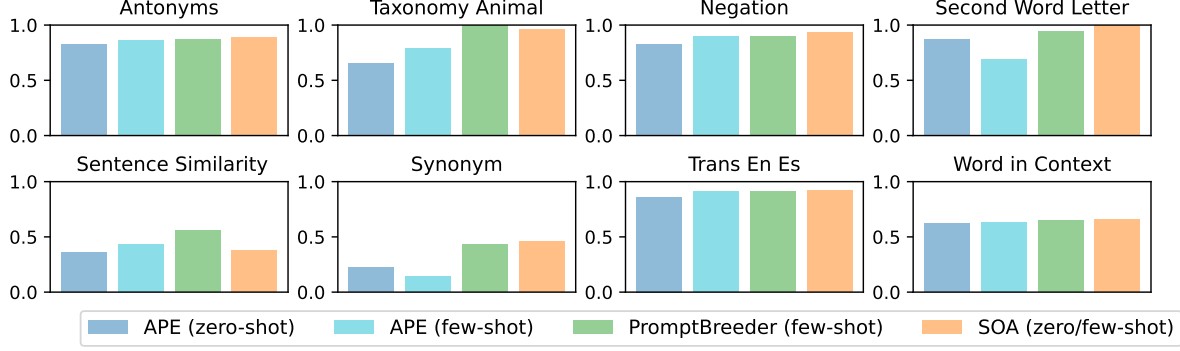

Figure 4: Test accuracy of SOA on the instruction induction tasks.

### 4.3 Analysis

**Applicability of SOA framework.** To study the general applicability of SOA framework, we conduct end-to-end optimization tasks on a variety of models covering both open-source LLMs and closed-source LLMs (API calls). As shown in Table 4, three end-to-end runs are implemented per task with the average performance and standard deviation reported. All experiments are initialized with SOA-pair method. We observed that GPT-4 performs the best in all tasks, followed by Llama3-70B. Claude 2 is comparable to GPT-3.5. For open-source LLM models, Mistral-7B and Llama3-8B are comparable to each other, both outperforming Llama2-7B by a large margin.

**Effect of Examples.** To compare with more latest work like OPRO and Evoprompt and better understand whether the performance gain introduced by SOA is largely caused by adding few-shot examples, we conduct

Table 4: `SOA` performance with different LLM models

| Method | Dis-ambiguation | Formal Fallacies | Hyperbaton | Salient Translation |
|---|---|---|---|---|
| GPT-3.5 | $69.99_{(2.95)}$ | $58.49_{(0.33)}$ | $84.35_{(1.83)}$ | $48.39_{(0.66)}$ |
| GPT-4 | $79.34_{(3.33)}$ | $75.91_{(0.53)}$ | $90.58_{(1.39)}$ | $70.45_{(0.99)}$ |
| PaLM 2 | $71.49_{(0.37)}$ | $58.33_{(1.53)}$ | $79.45_{(0.98)}$ | $49.07_{(3.25)}$ |
| Claude 2 | $72.95_{(2.26)}$ | $49.46_{(1.52)}$ | $83.32_{(1.01)}$ | $61.82_{(0.38)}$ |
| Mistral-7B | $65.89_{(0.76)}$ | $53.23_{(1.74)}$ | $78.76_{(1.36)}$ | $43.84_{(1.00)}$ |
| Llama2-7B | $42.74_{(4.61)}$ | $56.72_{(1.37)}$ | $53.23_{(2.37)}$ | $21.23_{(1.01)}$ |
| Llama3-8B | $62.63_{(3.85)}$ | $71.50_{(4.85)}$ | $57.52_{(4.28)}$ | $37.09_{(2.86)}$ |
| Llama3-70B | $74.73_{(2.01)}$ | $70.93_{(2.25)}$ | $82.26_{(0.66)}$ | $62.90_{(1.97)}$ |

an experiment by randomly adding two few-shot examples to OPRO and EvoPrompt. Our results, as shown in Table 5, indicated that OPRO exhibited a performance gain on only one out of four tasks while EvoPrompt showed improvement in two out of four tasks. This suggests the need for caution as performance degrades if optimized instructions do not align with naive few-shot selection.

Looking at the best prompt generated by `SOA` in Appendix G, we noted that 4 out of 8 of the optimal prompts for tasks in Table 2 did not contain any few-shot examples. This observation suggests that `SOA`'s ability to arrive at the most effective prompt does not depend on whether or not there are few-shot examples available. `SOA` truly optimizes the prompt based on the specific task at hand.

Table 5: Effect of few-shot (fs) examples on 4 BBH tasks.

| Method | Dis-ambiguation | Formal Fallacies | Hyperbaton | Salient Translation |
|---|---|---|---|---|
| OPRO | 71.53 | 49.51 | 75.92 | 43.88 |
| OPRO-few-shot | 66.93 | 52.41 | 62.90 | 37.39 |
| EvoPrompt | 53.70 | 50.81 | 74.79 | 47.58 |
| EvoPrompt-few-shot | 57.43 | 43.54 | 79.83 | 31.45 |
| AELP | 64.70 | 58.25 | 53.74 | 41.43 |
| `SOA`-pair | **72.37** | **58.87** | 86.02 | **48.19** |
| `SOA`-example | 68.47 | 58.65 | **87.51** | 47.59 |

**Effect of Hamming Distance.** We examine the impact of hamming distance with the performance-based vectors in comparison to cosine distance and lexical embedding for similarity calculation. The study encompasses both approaches carried out in 4 iterations using the same initial pool. Table 6 displays the outcomes on four BBH tasks. The results indicate that the hamming distance with performance-based vectors outperforms the cosine distance with lexical embedding, showing higher average and maximum scores, particularly for Disambiguation (+5.2) and Hyperbaton (+4.6) tasks.

Table 6: Effect of performance vector with hamming distance compared to cosine similarity.

| Method | Causal Judgement | | Disambiguation | | Hyperbaton | | Salient Translation | |
|---|---|---|---|---|---|---|---|---|
| | Average score | High score | Average score | High score | Average score | High score | Average score | High score |
| Cosine distance | $64.70_{(2.31)}$ | $67.86_{(2.47)}$ | $58.96_{(1.47)}$ | $63.30_{(0.00)}$ | $74.70_{(1.60)}$ | $85.7_{(0.00)}$ | $49.56_{(1.07)}$ | $\mathbf{58.80}_{(0.00)}$ |
| Hamming distance | $\mathbf{65.74}_{(2.87)}$ | $\mathbf{69.60}_{(2.97)}$ | $\mathbf{64.11}_{(1.28)}$ | $\mathbf{66.94}_{(2.88)}$ | $\mathbf{79.30}_{(4.48)}$ | $\mathbf{86.78}_{(2.15)}$ | $\mathbf{50.33}_{(2.32)}$ | $\mathbf{58.80}_{(0.00)}$ |

**Initialization Strategy.** The `SOA` can accommodate two types of inputs: *input/output pair* (`SOA`-pair) and *prompt example* (`SOA`-example), each bringing its own benefits. When using the *input/output pair* approach, the initialization occurs solely based on LLM's proposal, resulting in greater diversity in the initial pool. On the other hand, *prompt example* empowers users to introduce prior knowledge without leaning on LLM interpretation, and consequently, it performs better in more complex tasks such as Dyck Languages, Logical Five, and Color Reasoning, as illustrated in Table 2.

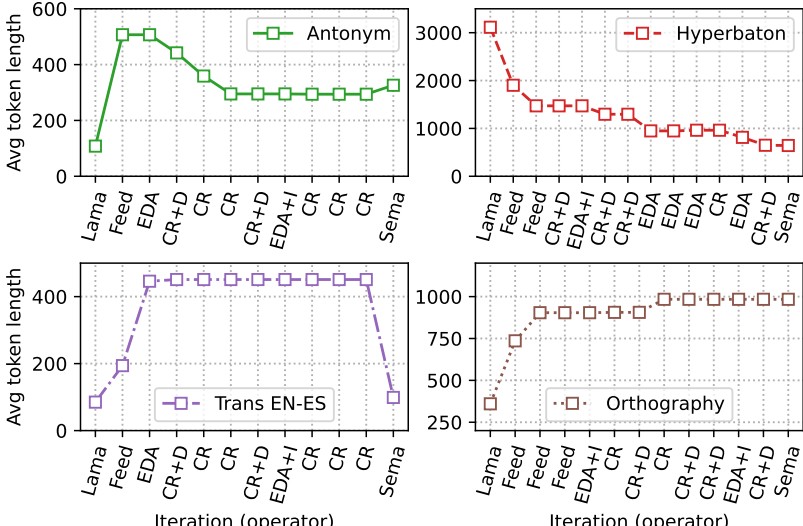

Figure 5: Variation of prompt length during `SOA` optimization. The prompt can be varied from zero-shot to few-shot, and few-shot to zero-shot. The operators are adaptively selected in each iteration.

**Effect of Operators on Prompt Length.** `SOA` aims to explore the entirety of the prompt space, spanning both zero-shot and few-shot scenarios. Understanding the variation in prompt length and the impact of the operator on this fluctuation is crucial. Fig. 5 provides a visual representation of the average prompt token length throughout the iterations. Interestingly, the length can either increase, decrease, or oscillate, which aligns with the "unfettered" expectations of global search. Specifically, we observed the initialization phase had a significant impact on prompt length. This observation is in agreement with our analysis of the Lamarckian and Feedback operators, which hold the power to both add and remove examples.

**Synthetic Few-shot Examples.** We observe that in certain cases `SOA` would generate novel synthetic few-shot examples instead of selecting from existing ones. To verify their veracity, we conduct a manual evaluation of the accuracy of the few-shot examples generated by `SOA` on a total of 24 instruction deduction tasks. We find that 90 out of the 92 examples evaluated (97.8%) are accurate. Among them, 24 out of the 92 (24.09%) are aligned with samples present in the training set. There are two cases where the synthetic example is inaccurate: the sentiment of *"A non-mystery mystery"* is identified as *"neutral"* where the ground truth is *"negative"*, and *"Little more than a well-mounted history lesson"* is identified as *"neutral"* where the ground truth is *"negative"*.

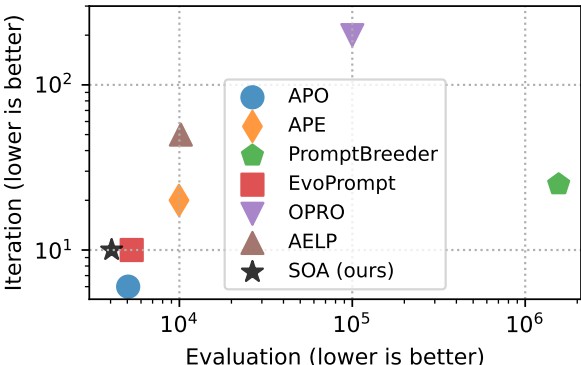

Figure 6: Comparison of computational cost on a total number of evaluations (x-axis) and iterations (y-axis).

**Computational Cost.** We monitor the computational cost of `SOA` based on two criteria: the number of model API calls consumed by evaluation and operator application, and the number of iterations. As shown in Fig. 6, `SOA` is the most cost-effective method that significantly reduces multiple orders of magnitude compared to evolution strategies, such as PromptBreeder. `SOA` also performs competitively in terms of iterations compared to the gradient descent approach, such as APO.

## 5 Related Work

In-context prompting is an efficient approach for communicating LLMs but the performance is strongly affected by the design of the prompt in specified tasks. Prompt Tuning to find the optimal prompt has

thus obtained broader attention. One research direction is the continuous prompt approaches that traverse the embedding space of input tokens to generate better prompts (Li & Liang, 2021; Zhang et al., 2021; Sun et al., 2022b;a; Chen et al., 2023). However, the optimized "soft" prompts from this paradigm often fall short of interpretability and are inaccessible for blackbox APIs. Discrete prompt approaches (Diao et al., 2022; Prasad et al., 2022), operating discrete tokens directly, offer an interactive interface to humans with better interpretability and show promising performance in various NLP tasks. Various methods have been proposed via gradient-based search (Shin et al., 2020), reinforcement learning (Zhang et al., 2022a; Deng et al., 2022; Sun et al., 2023) and ensemble methods (Hou et al., 2023; Pitis et al., 2023) while these methods encounter concerns in terms of scalability, reliability and efficiency (Wang et al., 2023).

Recent advancements rely on iterative sampling, scoring, and selection of exceptionally promising prompts, generating diverse possibilities for prompt optimization (Fernando et al., 2023; Guo et al., 2023; Hsieh et al., 2023), which proposed leveraging LLMs to implement evolution strategies in prompt searches. Yang et al. (2023a) demonstrates the capability of LLM as optimizers in prompt design. Pryzant et al. (2023) and Zhou et al. (2023) utilize natural language feedback to refine prompt instructions. However, these prompt evolution/refinement strategies largely focus on prompt instructions, typically short sentences. Our research reformulates the problem by permitting unrestrained tuning of a jointly in-context prompt, incorporating both instructions and examples, and offering more avenues for improvement, yet it also poses new challenges in navigating the high-dimensional combined space, while retaining high efficiency. While previous search and sampling algorithms have been investigated, such as Monte Carlo search (Zhou et al., 2023), Gibbs sampling (Xu et al., 2023), Beam search (Pryzant et al., 2023), or Evolution Algorithm (Fernando et al., 2023), we introduce a novel dual exploration-exploitation strategy that leverages the in-depth traits of each operator, utilizing an intuitive blend of global-local search, conducive to enhancing interactive dynamics during optimization.

## 6 Conclusion and Discussion

In this work, we propose a joint prompt-tuning framework that enables the combined optimization of prompt instruction and examples for LLMs. Benefiting from the global-local phased strategy and the adaptive selection of operators, SOA achieves state-of-the-art performance over a wide range of benchmark tasks while significantly reducing the computational cost. Despite having such achievements, SOA still needs thousands of inference calls in several iterations, which might be insufficient for supporting large-scale applications. Future work could explore better online strategies to further improve efficiency, and also investigate multi-objective prompt tuning beyond single accuracy or performance metric, such as safety, security, and robustness.

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

## A  Appendix

## B  Operator Definition

Operators are used to generate new candidates. Seven types of operators, broadly categorized into five classes are used by `SOA`. The idea is to provide a diverse set of operators so that a broad cognitive space of linguistics is covered.

### B.1  Lamarckian Operator

The Lamarckian operator follows the principles proposed in APE and Prompt Breeder (Zhou et al., 2023; Fernando et al., 2023). Given a set of input-output pairs for the task, an LLM agent is used to reverse-engineer the prompt from the provided demonstrating pairs. This type of operator allows a diverse set of prompt candidates to be generated with no prior knowledge of the task. Any prompt candidate will have to be induced from the demonstrating pairs. The prompt used by the LLM agent is in Table 11.

(Lamarckian Operator) Given a set of input/output pairs $(\mathcal{Q}, \mathcal{A}) = [(Q_1, A_1), ..., (Q_m, A_m)]$ and a base LLM $\mathcal{L}$, the Lamarckian operator is to reverse engineer the instruction $\mathcal{O}_L$ so that $\mathcal{O}_L(Q_i) = A_i, i = 1, ..., m$.

### B.2  Feedback Operator

Inspired by the concept of *Gradient Descent* in machine learning model training, we introduce an LLM agent that works as an examiner which examines the cases where the current task prompt fails and provides improvement guidance. Such guidance will be treated as *gradient* and be used by another LLM Agent as an improver to generate a new candidate. Though similar to what is proposed in APO (Pryzant et al., 2023), instead of only using gradient descent repeatedly, which has a higher probability of arriving at a local minimum, we take advantage of its fast converge rate to local minimum and combine it with other operators to target global minimum. When applying the Feedback operator, it will be applied to every candidate in the current pool. The prompt can be found in Table 12 - 13.

(Feedback Operator) The Feedback operator generates a new prompt $p'$ based on the existing prompt $p \in \mathcal{P}$, and where $p$ made mistakes for a task. The feedback operator $\mathcal{O}_F$ first looks at the cases where the current $p$ failed to generate a list of advice $G$, and then asks LLM $\mathcal{L}$ to apply such advice $G$ to existing prompt $p$ for generating the new prompt $p'$.

### B.3  Estimation of Distribution Operator

The next class of operators takes a set of parents as input to generate a modified candidate.

**Estimation of Distribution Operator (EDA)**: Following the principles proposed by (Hauschild & Pelikan, 2011) and work in (Fernando et al., 2023), we use a LLM agent that is fed with a subset of the current pool to generate new candidate. To ensure the diversity and quality of the subset, we first rank the candidates in the current pool by their performance in descending order. Then starting from the first item in the ordered candidates, we only add the candidate to the subset if it does not have a similarity score over a threshold with any other candidate that is already in the subset. This way candidates with higher performance are more prone to be added to the subset and the diversity of the subset is achieved. More details on how similarity is calculated can be found in section 3.3. The subset will be randomized before feeding into the LLM agent so the candidate's performance does not dictate its order. The prompt can be found in Table 14.

**EDA and Index Operator**: This is a variant of the EDA operator above. Based on the observations that LLM is more prone to use examples that appear late in the in-context learning (Liu et al., 2023; Fernando et al., 2023), after generating the subset following procedures of EDA, the subset is ordered by their performance in *ascending order*. To further balance exploitation and exploration and avoid being too biased over the candidate with the highest performance (Fernando et al., 2023), we instructed LLM that the candidates are ranked by their performance in *descending order* so that the low performance candidates are taken into consideration. The prompt can be found in Table 15.

(Estimation of Distribution Operator - EDA) EDA generates a new candidate based on a list of parents. It is a function operator $\mathcal{O}_E$ that performs $\mathcal{O}_E(\mathcal{P}, \mathcal{L}) = p'$. Given a list of prompts $\mathcal{P} = [p_1, ..., p_m]$ and an LLM $\mathcal{L}$, EDA provides a new prompt $p'$. Items in $\mathcal{P}$ satisfy the restriction that $d(p_i, p_j) < t$, where $d$ is a function that calculates similarity, and $t$ is a predefined threshold. If the items in $\mathcal{P}$ are ordered based on certain criteria, we call it EDA + Index (EDA+I).

## B.4    Crossover Operator

This class of operators takes two parents as input to generate a crossover candidate. The prompt can be found in Table 16.

**Crossover Operator(CR)**: Following the concept of crossover in the evolution algorithm, we introduce an LLM agent to function as a crossover operator that takes two parents and generates a crossover candidate. It takes the best two candidates in the current pool, namely the top two candidates with the highest performance, and performs linguistic crossover.

**Crossover with Diversity Operator(CR+D)**: This is a variance of the Crossover Operator. To provoke exploration, we follow a similar process in EDA where diversity in parents is considered. Thus it takes the best candidate and the most distinct individual to it as two parents for crossover operation. The distinctness between two candidates is measured by a similarity score. More details on how the similarity score is calculated can be found in section 3.3.

(Crossover Operator - CR) Crossover generates a new candidate based on two parents. It is a function operator $\mathcal{O}_C$ that performs $\mathcal{O}_C(p_1, p_2, \mathcal{L}) = p'$ where $p_1, p_2$ are two prompts selected from a prompt pool $\mathcal{P}$ where $\mathcal{P} = [p_1..., p_m]$, $p'$ is the generated prompt that hold features from both $p_1$ and $p_2$. If $p_2 = \arg\min_{p \in \mathcal{P}} d(p_1, p_i)$ is applied for choosing $p_2$, we call it Crossover + Distinct (CR + D).

## B.5    Semantic Operator

This class of operators takes a candidate and uses an LLM agent to compose a new candidate that shares its semantic meaning. When applying the Semantic operator, it will be applied to every candidate in the current pool. The prompt can be found in Table 17.

(Semantic Operator) The Semantic operator is a function operator $\mathcal{O}_S$ that performs $\mathcal{O}_S(p, \mathcal{L}) = p'$ where $p'$ is the generated prompt that shares the same semantic meaning as $p$.

# C Operator Analysis

## C.1 Few-Shot Additional / Removal Analysis

The ability to add and remove few-shot examples is crucial in dictating whether it is possible to traverse the full in-context prompt space. Thus we conduct the few-shot addition/ removal analysis to gauge each operator's capabilities. Below are examples of how operators add or remove few-shot examples.

Table 7: Lamarckian Operator Add Few-shot Example

| | |
|---|---|
| **Operator Input** | I gave a friend an instruction and some examples The friend read the instruction and wrote an output for every one of the inputs.
Here are the input-output pairs:

## Example ##
## Input ## : 92 24
## Output ##: ['68']

The instruction was: |
| **Operator Output** | Subtract the second number from the first number and write the result. If the result is negative, write the absolute value of the result.

Here are the input-output pairs:

Input: 10 5
Output: ['5']

Input: 50 75
Output: ['25']

Input: 100 100
Output: ['0']

Input: 7 20
Output: ['13']

Input: 0 5
Output: ['5']

Input: -10 5
Output: ['15']

Input: 92 24
Output: ['68']

Input: 50 -75
Output: ['125'] |

Table 8: Feedback Operator Add Few-shot Example: In this operation, few-shot examples are added based on the feedback. Individual feedback and their corresponding changes are colorcoded.

| **Original Prompt** | For each word, find its antonym and write it as the output. |
|---|---|
| **Generated Feedback** | 1. Clarify the definition of "antonym" to ensure that the prompt is asking for the opposite meaning of the word, rather than just a different word that is related in some way.

2. Provide more specific instructions for determining the antonym, such as indicating whether the opposite meaning should be an adjective or a noun.

3. Add a list of common antonyms for each word to help users who may not be familiar with all possible opposites.

4. Provide a context or sentence for each word to help users determine the appropriate antonym. |
| **Generated Prompt** | For each word, find its antonym, which is a word that has the opposite meaning, and write it as the output. The antonym should be a word that is commonly used to express the opposite meaning of the given word, and it should be of the same part of speech (e.g. noun, verb, adjective, adverb).

To help you identify the antonym, we have provided a list of common antonyms for each word. Additionally, we have included a context or sentence for each word to help you determine the appropriate antonym.

Example:
Word: Hot
Antonym: Cold
Context: The soup is too hot to eat.

Please note that some words may have multiple antonyms depending on the context, so use your best judgment to determine the most appropriate one. |

Table 9: Feedback Operator Remove Few-shot Example: In this operation, few-shot examples are removed based on the feedback. Individual feedback and their corresponding changes are colorcoded.

| | |
|---|---|
| **Original Prompt** | For each word, remove the prefix or suffix that indicates a negative or opposite meaning, and write the resulting word as the output.

For example, for the input "destabilize," remove the prefix "de-" which indicates a negative meaning, and the resulting word is "stabilize |
| **Generated Feedback** | 1.Clarify the prompt to specify that it only applies to words with a negative or opposite prefix or suffix. This will prevent confusion and mistakes when dealing with words that do not have such affixes.

2. Provide examples of common negative or opposite prefixes and suffixes, such as "un-" or "-less," to help users identify them more easily.

3. Include a check to ensure that the resulting word is a valid English word, to avoid outputs like "ane" or "direct" that are not actual words. This could be done by cross-referencing the output with a dictionary or word list.

4. Consider adding an option to retain the original word if no negative or opposite prefix or suffix is present. This would eliminate confusion when dealing with words like "linear" that do not have such affixes. |
| **Generated Prompt** | For each word that contains a negative or opposite prefix or suffix, remove the affix and write the resulting word as the output. Common negative or opposite prefixes and suffixes include "un-," "dis-," "in-," "-less," and "-ful." Please note that this prompt only applies to words with these specific affixes.

To ensure that the resulting word is a valid English word, we will cross-reference the output with a dictionary or word list. If the resulting word is not a valid English word, it will be excluded from the output.

If the original word does not contain a negative or opposite prefix or suffix, you may choose to retain the original word or exclude it from the output. Please specify your preference in the input. |

Table 10: Semantic Operator Remove Few-shot Example

| | |
|---|---|
| **Operator Input** | Order adjectives correctly in English sentences.

Q: Which sentence has the correct adjective order:
Options:
(A) rubber terrible ship
(B) terrible rubber ship
A: Let's think step by step.
When there is more than one adjective before a noun, the adjectives need to respect the following order before a noun: "[1. opinion] [2. size] [3. age] [4. shape] [5. color] [6. origin] [7. material] [8. purpose] noun". Option (A): "rubber terrible ship". (1) rubber" falls into the material category. (2) "terrible" falls into the opinion category. Option (A) has the following adjective order: [7. material] [1. opinion] (or, in numeric terms, 7 1). Because 7 < 1 is not correct, (A) does not have the correct ordering. Option (B): "terrible rubber ship". Option (B) has the following adjective order: [1. opinion] [7. material] (or, in numeric terms, 1 7). Because 1 < 7 is correct, (B) has the correct ordering. So the answer is (B).

Q: Which sentence has the correct adjective order:
Options:
(A) repulsive small Brazilian exercise ship
(B) Brazilian repulsive exercise small ship
A: Let's think step by step.
When there is more than one adjective before a noun, the adjectives need to respect the following order before a noun: "[1. opinion] [2. size] [3. age] [4. shape] [5. color] [6. origin] [7. material] [8. purpose] noun". Option (A): "repulsive small Brazilian exercise ship". (1) "repulsive" falls into the opinion category. (2) "small" falls into the size category. (3) "Brazilian" falls into the origin category. (4) "exercise" falls into the purpose category. Option (A) has the following adjective order: [1. opinion] [2. size] [6. origin] [8. purpose] (or, in numeric terms, 1 2 6 8). Because 1 < 2 < 6 < 8 is correct, (A) has the correct ordering. Option (B): "Brazilian repulsive exercise small ship". Option (B) has the following adjective order: [6. origin] [1. opinion] [8. purpose] [2. size] (or, in numeric terms, 6 1 8 2). Because 6 < 1 < 8 < 2 is not correct, (B) does not have the correct ordering. So the answer is (A).

Q: Which sentence has the correct adjective order:
Options:
(A) blue gold wonderful square shoe
(B) wonderful square blue gold shoe
A: Let's think step by step.
When there is more than one adjective before a noun, the adjectives need to respect the following order before a noun: "[1. opinion] [2. size] [3. age] [4. shape] [5. color] [6. origin] [7. material] [8. purpose] noun". Option (A): "blue gold wonderful square shoe". (1) "blue" falls into the color category. (2) "gold" falls into the material category. (3) "wonderful" falls into the opinion category. (4) "square" falls into the shape category. The adjective order that Option (A) has is [5. color] [7. material] [1. opinion] [4. shape] (or, in numeric terms, 5 7 1 4). Because 5 < 7 < 1 < 4 is not correct, (A) does not have the correct ordering. Option (B): "wonderful square blue gold shoe". Option (B) has the following adjective order: [1. opinion] [4. shape] [5. color] [7. material] (or, in numeric terms, 1 4 5 7 ). Because 1 < 4 < 5 < 7 is correct, (B) has the correct ordering. So the answer is (B). |
| **Operator Output** | Rearrange the adjectives in the given sentence in the correct order. |

### C.2 Operator Feature Analysis

To study the features of each operator we conduct a preliminary experiment where we study four operators: EDA Operator, Crossover, Feedback Operator, and Semantic Operator.

**Initialization:** As the initialized points have a tremendous impact on optimization problems. We randomly use four different seeds to create four initial pools for four different tasks: Causal Judgement, Salient Translation Error Detection, Disambiguation QA, and Hyperbaton. The idea is to provide various initialization points so that the performance of operators can be averaged to rule out the influence of initialization.

**Operator Applications:** For each initialization, we apply the following procedure for all four operators.

- For one round, starting with the initial pool, we consecutively apply the operator 5 times. This is to study the value of applying the operator consecutively.

    - For EDA and CrossOver, as they require multiple parents, we keep a pool size of 5 for each iteration after applying the operator. Performance gain is defined as whether the average performance of the pool is improved.
    - For Feedback Operator and Semantic Operator, as they only need one parent, we apply them to a random candidate from the initial pool and use the new candidate as the base for the next round. Performance gain is defined as whether the new candidate has a higher performance than its parent.

- To reduce the impact of randomness, we run this process 5 rounds for each operator.

Thus for each operator, it will be run a total of 4 tasks * 5 rounds * 5 application = 100 times.

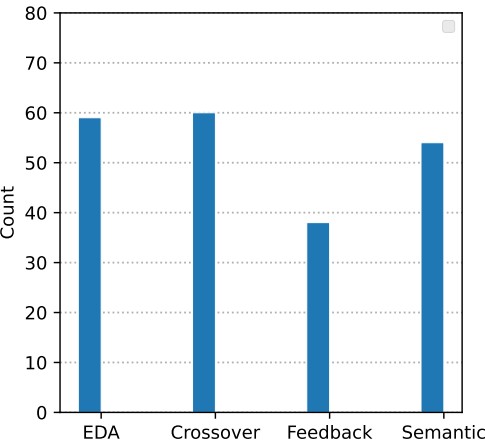

Figure 7: Operator Improvement Count

**Analysis:** There are two aspects we are particularly interested in. The first is **what the likelihood of performance gain when applying an operator is** (Probability of Improvement), and the second is **how fast each operator can continuously bring improvement** (Convergence Speed).

- **Probability Of Improvement**: Figure 7 shows the number of times performance is improved by each operator. Crossover and EDA Operator introduces improvements in more steps with Semantic Operator ranking third. Feedback Operator introduces the least number of improvements. This result helps populate the *Prob* column in table 1.

- **Convergence Speed**: Figure 8 shows that for each operator, as they are applied in 5 consecutive steps, the number of times improvement is introduced for each step. Figure 9 shows the average percentage of performance gain operators brought in each step.

- For EDA Operator and Crossover, each 5 step has a similar number of contributions for performance gains as shown in figure 8. From figure 9 we can also observe the first step brings the most improvement and the first 4 steps bring a similar improvement ratio.
- For Feedback Operator and Semantic Operator, the first step has a significantly higher chance of introducing improvement as shown in figure 8. This is especially true for Feedback Operator where step 1 accounts for over 34% of the total improvement counts. As for the improvement ratio, the first step for both Feedback Operator and Semantic Operator introduces significantly more improvements than the rest of the steps shown in figure 9.

Based on the tests, we learned that the value gained for applying Feedback Operator and Semantic Operator is significantly reduced after the 1st application. We interpret it as **Feedback Operator and Semantic Operator can jump to the local minimum pretty fast**, namely in 1 step, thus leading to less possibility of improvement for steps 2 - 5. Whereas for EDA Operator and Crossover, as they are merging genetic information between candidates, the likelihood of improvement is relatively randomized. So even if the first round of applying them renders no improvement, there is still a chance of performance gain in the following run. In other words, **we should be more patient with EDA Operator and Crossover**. Thus the operator tolerance (described in section 3.3-design 2) for EDA and Crossover is set to 4 and for Feedback Operator and Semantic Operator is 1. These learnings help populate the *Speed* column in table 1.

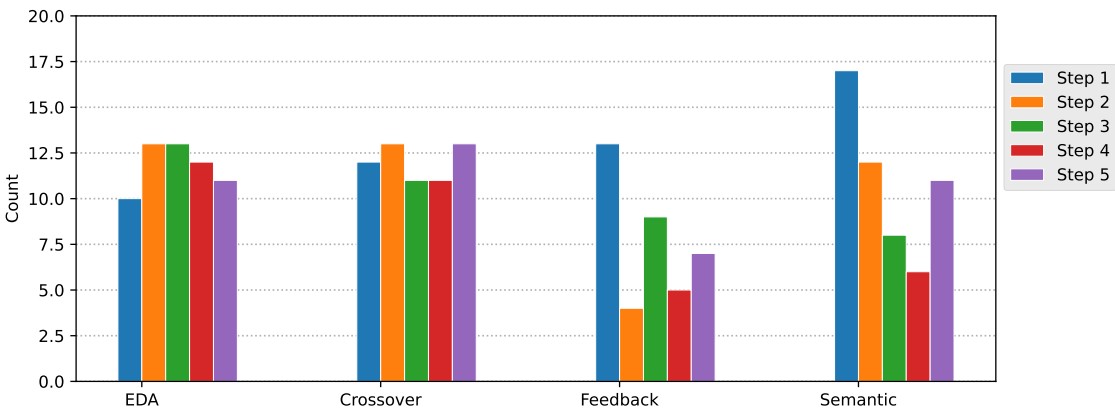

Figure 8: Operator Improvement Pattern: EDA Operator and Crossover have similar improvement counts for each step whereas for Feedback Operator and Semantic Operator, the first step introduced significantly more times of improvement compared to the others.

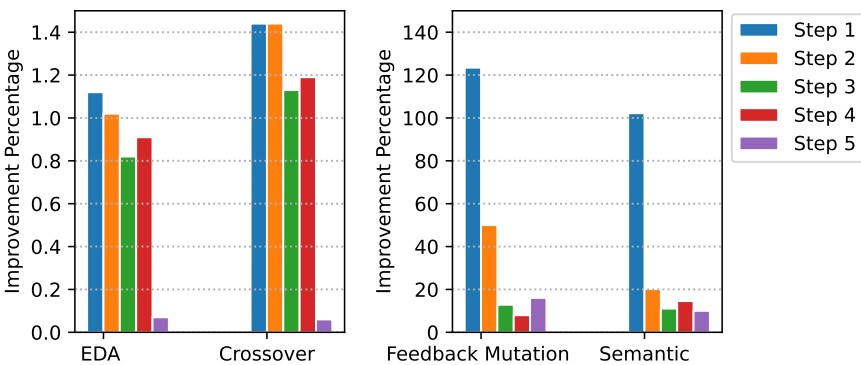

Figure 9: Improvement Ratio: On the left, for EDA and Crossover, we observe an almost equal improvement ratio for the first four steps. Improvement Ratio is defined as the relative percentage of improvement in the average performance for the entire pool. On the right, for Feedback and Semantic Operator, we observe the first round contributes significantly more improvement compared to the others. As Feedback and Semantic Operators take one input candidate, Improvement Ratio is defined as the relative performance improvement percentage for the candidate after applying the operator.

# D   Operator Prompts

**Operator Implementation:**   The state-of-art frameworks such as APO, EVOPROMPT, and AELP have already implemented operators such as feedback operator, crossover operator, and semantic operator with LLM. However, these implementations inflict restrictions on LLM with prompts. For example, in APO when implementing the feedback operator, the prompt specifically identified the use case to be zero-shot. (Pryzant et al., 2023) In EVOPROMPT-DE, when applying crossover operators, the focus is to only change the parts that two parents differentiate from each other. (Guo et al., 2023) In AELP, when applying semantic operators, it is restricted to a sentence level, not the whole prompt. (Hsieh et al., 2023). In `SOA`, we pay special attention not to apply any restrictions in our operator prompt, realizing the full potential of LLMs.

Table 11: Lamarckian Operator Prompt

---

I gave a friend an instruction and some input. The friend read the instruction and wrote an output for every one of the inputs. Here are the input-output pairs:

## Example ##
{*input output pairs*}

The instruction was:

---

Table 12: Gradient Descent Generation Prompt: Unlike APO which is also using gradient descent, we are NOT adding restrictions such as *"zero-shot classifier prompt."*, nor providing any differentiation between *instructions* and *examples*. Instead, we specifically ask LLM to output multiple feedback in one go. Also as are passing in the existing prompt as a whole, thus feedback should be on the paragraph/prompt level instead of the sentence/instruction level. We highlight the design that helps us achieve this below.

---

You are a quick improver. Given an existing prompt and a series of cases where it made mistakes. Look through each case carefully and identify what is causing the mistakes. Based on these observations, output ways to improve the prompts based on the mistakes.

## Existing Prompt ##
{*existing prompt*}

## Cases where it gets wrong:##
{*wrong cases*}

ways to improve the existing prompt based on observations of the mistakes in the cases above are:

---

Table 13: Gradient Descent Application Prompt: Following the principle of optimizing prompt as a whole, our operator prompts take input and output on the entire prompt level

---

You are a quick improver. Given an existing prompt and feedback on how it should improve. Create an improved version based on the feedback.

## Existing Prompt ##
{*existing prompt*}

## Feedback##
{*feedback*}

## Improved Prompt##

---

Table 14: EDA Prompt

---

You are a mutator. Given a series of prompts, your task is to generate another prompt with the same semantic meaning and intentions.

## Existing Prompts ##
{*existing prompt*}

The newly mutated prompt is:

---

Table 15: EDA+Index Prompt: The difference between EDA + Index and EDA is that EDA + Index takes advantage of the in-context learning technique and informs the order of the passed-in prompts

---

You are a mutator. Given a series of prompts, your task is to generate another prompt with the same semantic meaning and intentions. The series of prompts are ranked by their quality from best to worst.

## Existing Prompts ##
{*existing prompt*}

The newly mutated prompt is:

---

Table 16: Cross Over Prompt

---

You are a mutator who is familiar with the concept of cross-over in genetic algorithm, namely combining the genetic information of two parents to generate new offspring. Given two parent prompts, you will perform a cross-over to generate an offspring prompt that covers the same semantic meaning as both parents.

# Example
Parent prompt 1: Now you are a categorizer, your mission is to ascertain the sentiment of the provided text, either favorable or unfavorable

Parent prompt 2: Assign a sentiment label to the given sentence from ['negative', 'positive'] and return only the label without any other text.

Offspring prompt: Your mission is to ascertain the sentiment of the provided text and assign a sentiment label from ['negative', 'positive'].

## Given ##
Parent prompt 1: {*prompt 1*}
Parent prompt 2: {*prompt 2*}
Offspring prompt:

---

Table 17: Semantic Operator Prompt: To provoke LLM's creativity, we do not restrict to the ==semantic== level but expand that to ==intentions==, allowing LLM to not ==stick to a sentence-by-sentence modification.==

---

You are a mutator. Given a prompt, your task is to generate another prompt with the ==same semantic meaning and intentions.==

# Example:
current prompt: Your mission is to ascertain the sentiment of the provided text and assign a sentiment label from ['negative', 'positive'].
mutated prompt: Determine the sentiment of the given sentence and assign a label from ['negative', 'positive'].

Given:
current prompt: {*existing prompt*}
mutated prompt::

---

# E    Details of Experiments

### E.1    Baselines

- **APE** (Zhou et al., 2023) uses LLM agent for instruction induction tasks. It proposes forward mode generation and reverse mode generation and uses log probability to generate and evaluate candidates.

- **APO** (Pryzant et al., 2023) uses feedback provided by LLM as gradients to approach prompt optimization. It uses beam search to find the best candidate.

- **PromptBreeder** (Fernando et al., 2023) uses the evolution algorithm to tackle prompt optimization tasks and utilizes thinking styles, and mutation prompts to surface the best task prompt.

- **AELP** (Hsieh et al., 2023) uses existing prompts (Suzgun et al., 2022b) to target long prompt optimization and improves them by mutating on a sentence level with history-guided search.

- **EVOPROMPT** (Guo et al., 2023) uses crossover mutation and semantic mutation with an evolution algorithm to find the best prompt.

- **OPRO** (Yang et al., 2023a) uses meta prompt, solution-score pairs, and task descriptions to generate candidates.

### E.2    Benchmark tasks

- **24 Instruction Induction Tasks**: These 24 instruction tasks (Honovich et al., 2022) span many facets of language understanding, from simple phrase structure to similarity and causality identification. Both training and testing data are provided for these tasks and we create our training and evaluation data set from the available training data and use the provided testing data set as is. Depending on the task, we use up to 50 training data and up to 50 evaluation data. We use *input/output pair* format for these tasks.

- **Ethos**: Ethos (Mollas et al., 2021) is an online English hate speech detection data set with 997 online comments and hate speech labels. We select 50 for training, 50 for evaluation, and 150 for testing. We use *prompt example* format for this data set following the practice of APO (Pryzant et al., 2023).

- **Liar**: Liar (Wang, 2017) is an English fake news detection data set with 4000 statements, context, and lie labels. We select 50 for training, 50 for evaluation, and 150 for testing. We use *prompt example* format for this data set following the practice of APO (Pryzant et al., 2023).

- **Sarcasm**: Sarcasm (Farha & Magdy, 2020) is an Arabic sarcasm detection data set with 10,000 online comments and sarcasm labels. We select 50 for training, 50 for evaluation, and 150 for testing. We use *prompt example* format for this data set following the practice of APO (Pryzant et al., 2023).

- **BBH**: BBH (Aarohi & bench authors, 2023) is a collaborative benchmark that aims to quantitatively measure the capabilities and limitations of language models. We followed the same practice in the AELP paper with the same tasks and randomly selected 50 for training, 50 for evaluation, and 125 for test. (Hsieh et al., 2023)

### E.3    `SOA` Setting

- **Pool Size**: In the experiments, for *phase 0: Global initialization* we set the pool size to be 15. For the rest phases, we set the pool to be 5.

- **Operator Tolerance**: Based on operator analysis in section C.2, the tolerance for Feedback Operator and Semantic Operator is set to 1. The tolerance for EDA Operator and Crossover is set to 4. Thus the minimum number of times operators will be applied in *phase 2: global evolution operation* is 8.

- **Model Configuration**: For operators, we set the temperature to 0.5 to tap into LLM's creativity. For evaluations, we set the temperature to 0.

- **Performance Gain in Stop Criteria**: To improve efficiency, when evaluating performance gain to decide whether we should move to the next phase, we are only looking at the best candidate in the current pool.

- **Candidate Selection**: To improve efficiency, after getting new candidates, we combine them with the current pool and use a greedy algorithm to select the top performer to be the new pool.

## F    Additional Experiment Results

### F.1    24 Instruction Induction Tasks

Table 18 shows the comparison between APE, PromptBreeder, and `SOA` evaluated by the best prompt on 24 instruction induction tasks. `SOA` outperforms 21/24 tasks over APE zero shot, 17 / 24 tasks over APE few shot and 18 / 24 tasks on Prompt Breeder. `SOA` generated few-shot prompts for 20 / 24 tasks and zero-shot examples for 4 / 24 tasks. For the full set of generated prompts please refer to table 26.

Table 18: 24 Instruction Induction Task in APE

| Task | APE (zero-shot) | APE (few-shot) | PromptBreeder (few-shot) | SOA-3.5 | SOA-4 |
|---|---|---|---|---|---|
| **Antonyms** | 0.83 | 0.86 | 0.87 | **0.89** | **0.91** |
| **Cause Effect** | 0.84 | 1 | 1 | 0.96 | **1** |
| **Common Concept** | 0.27 | **0.32** | 0 | 0.23 | 0.28 |
| **Diff** | 1 | 1 | 1 | **1** | **1** |
| **First Word Letter** | 1 | 1 | 1 | **1** | **1** |
| **Informal Formal** | 0.65 | **0.70** | 0.07 | 0.6 | 0.67 |
| **Large Animal** | 0.97 | 0.97 | **0.97** | 0.96 | 0.94 |
| **Letters List** | 0.99 | 1 | 0.99 | **1** | **1** |
| **Taxonomy Animal** | 0.66 | 0.79 | 1 | 0.96 | **1** |
| **Negation** | 0.83 | 0.9 | 0.9 | **0.94** | 0.88 |
| **Num Verb** | 1 | 1 | 1 | **1** | **1** |
| **Active Passive** | 1 | 1 | 1 | **1** | **1** |
| **Singular Plural** | 1 | 1 | 1 | **1** | **1** |
| **Rhymes** | 1 | 0.61 | 1 | **1** | **1** |
| **Second Word Letter** | 0.87 | 0.69 | 0.95 | **1** | **1** |
| **Sentence Similarity** | 0.36 | 0.43 | **0.56** | 0.38 | 0.55 |

Continuation of Table 18

| Continuation of Table 18 | | | | | |
|---|---|---|---|---|---|
| **Sentiment** | 0.94 | 0.93 | 0.93 | **0.94** | **0.94** |
| **Orthography Starts** | 0.68 | 0.69 | 0.71 | **0.72** | **0.94** |
| **Sum** | 1 | 1 | 1 | **1** | **1** |
| **Synonym** | 0.22 | 0.14 | 0.43 | **0.46** | 0.38 |
| **Trans En De** | 0.72 | 0.86 | 0.87 | 0.83 | **0.96** |
| **Trans En Es** | 0.86 | 0.91 | 0.91 | **0.92** | **0.94** |
| **Trans En Fr** | 0.78 | 0.9 | 0.91 | 0.88 | **0.93** |
| **Word in Context** | 0.62 | 0.63 | 0.65 | **0.66** | **0.7** |

### F.2 Generated Prompt Comparison

We notice that the **prompts generated by `SOA` are easier to understand by humans.** Below is a comparison between prompts generated for task Rhymes. The task description is: *"Write a word that rhymes with the input word."*

The prompt generated by APE and Instruct Zero does not fit the task. The prompt generated by Prompt Breeder is not easy to understand how it relates to rhyme. The prompt generated by `SOA` is easy to understand with few shot examples added.

Table 19: Generated Prompt Comparison for task "Rhymes"

| Framework | Generated Prompt |
|---|---|
| **APE** | write a function that takes in a string and outputs the string with the first letter capitalized. |
| **Instruct Zero** | Write a function that takes a word as input and returns the output word. |
| Continuation of Table 19 | |

Continuation of Table 19

| Prompt Breeder | Prompt 0: If the last letter of the input is 'e', remove it. |
|---|---|
| | Prompt 1: remove the last two letters of the input and add the letters \xc2 \x93mote \xc2 \x94. |
| | Contexts |
| | Context 0: |
| | Q. pea |
| | A. If the last letter of the input is 'e', remove it. |
| | A. If the last letter of the input is 's', remove it. |
| | A. If the last letter of the input is 'y', remove it. |
| | A. If the last letter of the input is |
| | remove the last two letters of the input and add the letters \xc2 \x93mote \xc2 \x94. |
| | Therefore, the correct answer is (a) pea. |
| | Context 1: |
| | Q. night |
| | A. If the last letter of the input is 'e', remove it. |
| | A. If the last letter of the input is 't', remove it. |
| | A. If the last letter of the input is 'h', remove it. |
| | A. If the last letter of the input is |
| | remove the last two letters of the input and add the letters \xc2 \x93mote \xc2 \x94. |
| | Therefore, the correct answer is (The answer is night. |
| | Context 2: |
| | Q. add |
| | A. If the last letter of the input is 'e', remove it. |
| | A. If the last letter of the input is 'd', remove it. |
| | A. If the last letter of the input is 'a', remove it. |
| | A. If the last letter of the input is |
| | remove the last two letters of the input and add the letters \xc2 \x93mote \xc2 \x94. |
| | Therefore, the correct answer is (The answer is added. |

Continuation of Table 19

| | Continuation of Table 19 |
|---|---|
| **SOA** | Create a list of words that rhyme with the given word. To ensure that your rhymes are accurate, make sure that the words have the same vowel sound and ending consonant sound. For example, "cat" rhymes with "bat" and "hat," but not with "dog" or "mat."

To get started, here are some examples of words that rhyme with the given word:

- Love: dove, glove, above, shove, of
- Time: rhyme, chime, climb, mime, prime

To find more rhyming words, you can use a rhyming dictionary, online resources, or brainstorm with friends. Be creative and try to use a variety of different rhyming words instead of repeating the same one multiple times.

To avoid common pitfalls, make sure to double-check your spelling and pronunciation of the words. Also, avoid using words that only partially rhyme or have a different stress pattern.

After you've created your list, ask for feedback on the quality of your rhymes. This can help you to improve and refine your skills.

For an added challenge, consider generating rhyming words that fit a particular theme or context. This can help you to focus your creativity and generate more interesting and relevant rhymes. |

# G Few-shot Add/ Removal Examples

Below are cases listing examples where few-shot examples are added or removed with `SOA`.

## G.1 Zero-shot to Few-shot

Table 20: Add Few-shot Example: added examples are highlighted.

| Task | Ethos |
|---|---|
| **Input Prompt** | Is the following text hate speech? \n ## Output Format ## True or False |
| **Output Prompt** | Your task is to evaluate whether the provided input includes any offensive language. This could be language that is sexist, racist, discriminatory, or advocates violence or harm. Offensive language can also be identified by an aggressive or belittling tone. However, remember that the mere presence of profanity does not necessarily make a statement offensive. The context and intention behind the statement should also be taken into account. If none of these conditions are fulfilled, return 'False'. 

 For instance, the phrase "You are a fool" is offensive due to its derogatory tone, so you should return 'True'. On the other hand, the phrase "This is a damn tasty pizza" includes profanity but is not offensive, so you should return 'False'. Similarly, a phrase like "All people of X race are lazy" is offensive because of its racist undertones, so you should return 'True'. In contrast, a phrase like "I dislike the color yellow" is not offensive, so you should return 'False'. |

## G.2 Zero-shot to Zero-shot

Table 21: Zero-shot to Zero-shot

| Task | Ethos |
|---|---|
| **Input Prompt** | Is the following text hate speech? \n ## Output Format ## True or False |
| **Output Prompt** | Classify the given text as hate speech or not and generate a binary output of 1 for Yes and 0 for No. |

## G.3 Few-shot to Zero-shot

Table 22: Few-shot to Zero-shot

| Task | Hyperbaton |
|---|---|
| Continuation of Table 22 | |

Continuation of Table 22

| **Input Prompt** | Order adjectives correctly in English sentences.

Q: Which sentence has the correct adjective order:
Options:
(A) rubber terrible ship
(B) terrible rubber ship
A: Let's think step by step.
When there is more than one adjective before a noun, the adjectives need to respect the following order before a noun: "[1. opinion] [2. size] [3. age] [4. shape] [5. color] [6. origin] [7. material] [8. purpose] noun".
Option (A): "rubber terrible ship". (1) rubber" falls into the material category. (2) "terrible" falls into the opinion category. Option (A) has the following adjective order: [7. material] [1. opinion] (or, in numeric terms, 7 1). Because $7 < 1$ is not correct, (A) does not have the correct ordering.
Option (B): "terrible rubber ship". Option (B) has the following adjective order: [1. opinion] [7. material] (or, in numeric terms, 1 7). Because $1 < 7$ is correct, (B) has the correct ordering. So the answer is (B).

Q: Which sentence has the correct adjective order:
Options:
(A) repulsive small Brazilian exercise ship
(B) Brazilian repulsive exercise small ship
A: Let's think step by step.
When there is more than one adjective before a noun, the adjectives need to respect the following order before a noun: "[1. opinion] [2. size] [3. age] [4. shape] [5. color] [6. origin] [7. material] [8. purpose] noun".
Option (A): "repulsive small Brazilian exercise ship". (1) "repulsive" falls into the opinion category. (2) "small" falls into the size category. (3) "Brazilian" falls into the origin category. (4) "exercise" falls into the purpose category. Option (A) has the following adjective order: [1. opinion] [2. size] [6. origin] [8. purpose] (or, in numeric terms, 1 2 6 8). Because $1 < 2 < 6 < 8$ is correct, (A) has the correct ordering.
Option (B): "Brazilian repulsive exercise small ship". Option (B) has the following adjective order: [6. origin] [1. opinion] [8. purpose] [2. size] (or, in numeric terms, 6 1 8 2). Because $6 < 1 < 8 < 2$ is not correct, (B) does not have the correct ordering. So the answer is (A).
... |
|---|---|
| **Output Prompt** | Identify the sentence with the correct order of adjectives: opinion, size, age, shape, color, origin, material, purpose. |

Table 23: Few-shot to Few-shot

| **Task** | Hyperbaton |
|---|---|
| Continuation of Table 23 | |

| | |
|---|---|
| | Continuation of Table 23 |

| | |
|---|---|
| **Input Prompt** | Order adjectives correctly in English sentences.

Q: Which sentence has the correct adjective order:
Options:
(A) rubber terrible ship
(B) terrible rubber ship
A: Let's think step by step.
When there is more than one adjective before a noun, the adjectives need to respect the following order before a noun: "[1. opinion] [2. size] [3. age] [4. shape] [5. color] [6. origin] [7. material] [8. purpose] noun".
Option (A): "rubber terrible ship". (1) rubber" falls into the material category. (2) "terrible" falls into the opinion category. Option (A) has the following adjective order: [7. material] [1. opinion] (or, in numeric terms, 7 1). Because $7 < 1$ is not correct, (A) does not have the correct ordering.
Option (B): "terrible rubber ship". Option (B) has the following adjective order: [1. opinion] [7. material] (or, in numeric terms, 1 7). Because $1 < 7$ is correct, (B) has the correct ordering. So the answer is (B).

Q: Which sentence has the correct adjective order:
Options:
(A) repulsive small Brazilian exercise ship
(B) Brazilian repulsive exercise small ship
A: Let's think step by step.
When there is more than one adjective before a noun, the adjectives need to respect the following order before a noun: "[1. opinion] [2. size] [3. age] [4. shape] [5. color] [6. origin] [7. material] [8. purpose] noun".
Option (A): "repulsive small Brazilian exercise ship". (1) "repulsive" falls into the opinion category. (2) "small" falls into the size category. (3) "Brazilian" falls into the origin category. (4) "exercise" falls into the purpose category. Option (A) has the following adjective order: [1. opinion] [2. size] [6. origin] [8. purpose] (or, in numeric terms, 1 2 6 8). Because $1 < 2 < 6 < 8$ is correct, (A) has the correct ordering.
Option (B): "Brazilian repulsive exercise small ship". Option (B) has the following adjective order: [6. origin] [1. opinion] [8. purpose] [2. size] (or, in numeric terms, 6 1 8 2). Because $6 < 1 < 8 < 2$ is not correct, (B) does not have the correct ordering. So the answer is (A).
... |

| | |
|---|---|
| | Continuation of Table 23 |

| | Continuation of Table 23 |
|---|---|
| **Output Prompt** | Adjective Order Practice:

In English grammar, the order of adjectives is important to convey accurate and effective descriptions. Here are two examples of sentences with adjectives. Determine which sentence has the correct adjective order.

Example 1:
a) The big, red, round ball bounced down the street.
b) The round, red, big ball bounced down the street.

Example 2:
a) The delicious, homemade, chocolate cake was devoured by the guests.
b) The chocolate, homemade, delicious cake was devoured by the guests.

Choose the letter of the sentence with the correct adjective order as your answer.

Note: Adjective order generally follows the pattern of opinion, size, age, shape, color, origin, material, and purpose. If the adjectives do not fit into this pattern, the order is determined by the speaker's preference.

If you choose the incorrect sentence, we will explain why it is wrong to help you learn from your mistakes. Good luck! |

## H Generated Prompts

In this section, we list the prompts generated by `SOA` with the best performance for each task. All prompts are generated by gpt-3.5. We observe a mix of few-shot prompts and zero-shot prompts for different tasks. This indicates both LLM's ability to perform in-context prompt optimization and `SOA`'s ability to traverse the whole problem space to find optimal solutions.

We also notice that the few-shot examples in the final prompts are largely generated by LLM instead of copied from example instruction or training sets. Thus it serves as further proof of LLM's capability of in-context prompt optimization and `SOA`'s credibility in this problem space.

Table 24: BBH Prompts

| | |
|---|---|
| **Causal Judgment** | Provide reactions to intentional actions in diverse scenarios, while also considering causation and its complexities. To assist with determining causation, provide specific guidelines and examples for each scenario. To avoid any confusion or misinterpretation, precise language and definitions will be used throughout the prompt. Additionally, feedback from experts and individuals with relevant experience in the field of causation will be incorporated to ensure accuracy and relevance. To challenge users' critical thinking skills, include diverse and complex scenarios that require creative problem-solving and a deeper understanding of causation in various areas of life. |
| **Dyke Languages** | Correctly close all brackets, including nested brackets, in the provided sequence in the proper order from innermost to outermost. Mistakes such as forgetting to close a bracket or closing brackets in the wrong order can result in an error. If an error is made, a clear and concise message will indicate which bracket is not properly closed and suggest how to correct it. A visual representation of the correct sequence of closed brackets is provided below: [ ( [ ( ) ] ) ] 

 Examples of valid and invalid inputs: 

 Valid input: [ ( ) ] 
 Valid input: [ ( [ ] ) ] 
 Invalid input: [ ( [ ) ] 
 Warning message: The bracket at position 8 is not properly closed. Please close the bracket to ensure proper syntax. 
 Suggested correction: [ ( [ ] ) ] 

 Invalid input: [ ( [ ] ) ] 
 Warning message: The bracket at position 8 is not properly closed. Please close the bracket to ensure proper syntax. 
 Suggested correction: [ ( [ ] ) ] |
| **Formal Fallacies** | Read the given argument carefully and determine whether it is deductively valid or invalid based on the explicitly stated premises. Provide a justification for your answer. |
| **Disambiguation QA** | For each sentence with a gender-neutral pronoun, determine the antecedent or state if it is ambiguous. Use (A) for the first option, (B) for the second option, or (C) for ambiguous. Additionally, provide an explanation of the antecedent (the person or thing the pronoun refers to) for each sentence. |

Continuation of Table 24

| Continuation of Table 24 | |
|---|---|
| **Hyperbaton** | Test your knowledge of adjective order in English sentences with interactive exercises and quizzes. Learn the rule of opinion-size-age-shape-color-origin-material-purpose noun and apply it to different types of nouns such as animals, objects, and people. Practice constructing your own sentences and receive feedback on incorrect answers to improve your skills. By the end of this exercise, you'll be able to confidently order adjectives and communicate accurately in English. |
| Continuation of Table 24 | |

Continuation of Table 24

| | On a plate, there are three fruits: a red apple, a yellow banana, and a green pear. The banana is positioned to the immediate left of the apple, meaning there are no other fruits between them. The pear is the rightmost fruit, meaning it comes last in the order.

Which of the following statements is true?

(A) The red apple is the leftmost fruit.
(B) The yellow banana is the leftmost fruit.
(C) The green pear is the leftmost fruit.

Explanation:
To solve this prompt, pay attention to the precise language used to describe the relationships between the fruits and their positions in the order. The banana is to the immediate left of the apple, meaning it is directly adjacent to it and there are no other fruits between them. The pear is the rightmost fruit, meaning it comes last in the order.

Therefore, the correct answer is (B) The yellow banana is the leftmost fruit.

To further practice this concept, here are some additional examples:

1. On a plate, there are three different colored balls: a blue ball, a red ball, and a green ball. The red ball is directly to the left of the blue ball, meaning there are no other balls between them. The green ball is the rightmost. Which of the following statements is true?
(A) The blue ball is the leftmost ball.
(B) The red ball is the leftmost ball.
(C) The green ball is the leftmost ball.

2. In a row of three houses, there is a blue house, a yellow house, and a green house. The blue house is in the middle, and the yellow house is directly to the left of the green house, meaning there are no other houses between them. Which of the following statements is true? (A) The blue house is the leftmost house.
(B) The yellow house is the leftmost house.
(C) The green house is the leftmost house.

Feedback:
If you selected the wrong answer, here's why:
(A) The red apple is not the leftmost fruit. The yellow banana is directly to its left.
(C) The green pear is not the leftmost fruit. The banana and the apple come before it in the order.

To avoid confusion, use precise language to describe the relationships between objects and their positions in the order. Avoid using vague terms like "newer" or "older" without specifying their exact relationship to other objects in the order. Provide more context or details to help clarify any ambiguities in the prompt. Make sure the order of the objects is clearly defined and consistent throughout the prompt. |
|:--|:--|
| **Logical Deduction Five** | |

Continuation of Table 24

Continuation of Table 24

| | |
|---|---|
| **Reasoning Colored Objects** | Identify the color of objects arranged in a row on a surface.
Q: On the desk, there is a black stapler, a green highlighter, a yellow ruler, a blue pen, and a purple marker. What color is the pen?
Options:
(A) red
(B) orange
(C) yellow
(D) green
(E) blue
(F) brown
(G) magenta
(H) fuchsia
(I) mauve
(J) teal
(K) turquoise
(L) burgundy
(M) silver
(N) gold
(O) black
(P) grey
(Q) purple
(R) pink
A: Let's think step by step.
According to this question, the objects are arranged in a row, from left to right, as follows: (1) a black stapler, (2) a green highlighter, (3) a yellow ruler, (4) a blue pen, and (5) a purple marker.
The pen is the fourth item on the list, namely (4). The color of the pen is blue. So the answer is (E). |
| **Salient Transla­tion Error Detection** | Read the following translations from German to English and identify the type of error present in each one. The error can be one of the following types: Named Entities, Numerical Values, Modifiers or Adjectives, Negation or Antonyms, Facts, or Dropped Content. Write the corresponding letter for each error type in the options provided.

For example:
Source: Der Hund ist braun.
Translation: The cat is brown.
The translation contains an error pertaining to:
Options:
(A) Modifiers or Adjectives
(B) Numerical Values
(C) Negation or Antonyms
(D) Named Entities
(E) Dropped Content
(F) Facts

Output: (D) |

Continuation of Table 24

Continuation of Table 24

| | |
|---|---|
| **Causal Judgment** | Provide reactions to intentional actions in diverse scenarios, while also considering causation and its complexities. To assist with determining causation, provide specific guidelines and examples for each scenario. To avoid any confusion or misinterpretation, precise language and definitions will be used throughout the prompt. Additionally, feedback from experts and individuals with relevant experience in the field of causation will be incorporated to ensure accuracy and relevance. To challenge users' critical thinking skills, include diverse and complex scenarios that require creative problem-solving and a deeper understanding of causation in various areas of life. |
| **Dyke Languages** | Correctly close all brackets, including nested brackets, in the provided sequence in the proper order from innermost to outermost. Mistakes such as forgetting to close a bracket or closing brackets in the wrong order can result in an error. If an error is made, a clear and concise message will indicate which bracket is not properly closed and suggest how to correct it. A visual representation of the correct sequence of closed brackets is provided below: [ ( [ ( ) ] ) ] 

 Examples of valid and invalid inputs: 

 Valid input: [ ( ) ] 
 Valid input: [ ( [ ] ) ] 
 Invalid input: [ ( [ ) ] 
 Warning message: The bracket at position 8 is not properly closed. Please close the bracket to ensure proper syntax. 
 Suggested correction: [ ( [ ] ) ] 

 Invalid input: [ ( [ ] ) ] 
 Warning message: The bracket at position 8 is not properly closed. Please close the bracket to ensure proper syntax. 
 Suggested correction: [ ( [ ] ) ] |
| **Formal Fallacies** | Read the given argument carefully and determine whether it is deductively valid or invalid b5rased on the explicitly stated premises. Provide a justification for your answer. |
| **Disambiguation QA** | For each sentence with a gender-neutral pronoun, determine the antecedent or state if it is ambiguous. Use (A) for the first option, (B) for the second option, or (C) for ambiguous. Additionally, provide an explanation of the antecedent (the person or thing the pronoun refers to) for each sentence. |
| **Hyperbaton** | Test your knowledge of adjective order in English sentences with interactive exercises and quizzes. Learn the rule of opinion-size-age-shape-color-origin-material-purpose noun and apply it to different types of nouns such as animals, objects, and people. Practice constructing your own sentences and receive feedback on incorrect answers to improve your skills. By the end of this exercise, you'll be able to confidently order adjectives and communicate accurately in English. |

Continuation of Table 24

Continuation of Table 24

|  |  |
|---|---|
| **Logical Deduction Five** | On a plate, there are three fruits: a red apple, a yellow banana, and a green pear. The banana is positioned to the immediate left of the apple, meaning there are no other fruits between them. The pear is the rightmost fruit, meaning it comes last in the order.

Which of the following statements is true?

(A) The red apple is the leftmost fruit.
(B) The yellow banana is the leftmost fruit.
(C) The green pear is the leftmost fruit.

Explanation:
To solve this prompt, pay attention to the precise language used to describe the relationships between the fruits and their positions in the order. The banana is to the immediate left of the apple, meaning it is directly adjacent to it and there are no other fruits between them. The pear is the rightmost fruit, meaning it comes last in the order.

Therefore, the correct answer is (B) The yellow banana is the leftmost fruit.

To further practice this concept, here are some additional examples:

1. On a plate, there are three different colored balls: a blue ball, a red ball, and a green ball. The red ball is directly to the left of the blue ball, meaning there are no other balls between them. The green ball is the rightmost. Which of the following statements is true?
(A) The blue ball is the leftmost ball.
(B) The red ball is the leftmost ball.
(C) The green ball is the leftmost ball.

2. In a row of three houses, there is a blue house, a yellow house, and a green house. The blue house is in the middle, and the yellow house is directly to the left of the green house, meaning there are no other houses between them. Which of the following statements is true? (A) The blue house is the leftmost house.
(B) The yellow house is the leftmost house.
(C) The green house is the leftmost house.

Feedback:
If you selected the wrong answer, here's why:
(A) The red apple is not the leftmost fruit. The yellow banana is directly to its left.
(C) The green pear is not the leftmost fruit. The banana and the apple come before it in the order.

To avoid confusion, use precise language to describe the relationships between objects and their positions in the order. Avoid using vague terms like "newer" or "older" without specifying their exact relationship to other objects in the order. Provide more context or details to help clarify any ambiguities in the prompt. Make sure the order of the objects is clearly defined and consistent throughout the prompt. |

Continuation of Table 24

Continuation of Table 24

| | |
|---|---|
| **Reasoning Colored Objects** | Identify the color of objects arranged in a row on a surface.

Q: On the desk, there is a black stapler, a green highlighter, a yellow ruler, a blue pen, and a purple marker. What color is the pen?

Options:
(A) red
(B) orange
(C) yellow
(D) green
(E) blue
(F) brown
(G) magenta
(H) fuchsia
(I) mauve
(J) teal
(K) turquoise
(L) burgundy
(M) silver
(N) gold
(O) black
(P) grey
(Q) purple
(R) pink

A: Let's think step by step.
According to this question, the objects are arranged in a row, from left to right, as follows: (1) a black stapler, (2) a green highlighter, (3) a yellow ruler, (4) a blue pen, and (5) a purple marker.
The pen is the fourth item on the list, namely (4). The color of the pen is blue. So the answer is (E). |

Continuation of Table 24

| Continuation of Table 24 | |
| --- | --- |
| **Salient Translation Error Detection** | Read the following translations from German to English and identify the type of error present in each one. The error can be one of the following types: Named Entities, Numerical Values, Modifiers or Adjectives, Negation or Antonyms, Facts, or Dropped Content. Write the corresponding letter for each error type in the options provided.

For example:
Source: Der Hund ist braun.
Translation: The cat is brown.
The translation contains an error pertaining to:

Options:
(A) Modifiers or Adjectives
(B) Numerical Values
(C) Negation or Antonyms
(D) Named Entities
(E) Dropped Content
(F) Facts

Output: (D) |

Table 25: APO Prompts

| Ethos | Does the provided text contain hate speech? Return a boolean value of True or False. |
|---|---|
| Liar | Analyze the context and other information provided to determine the truthfulness of the statement. To do so, consider the following guidelines:

1. Identify key sources of information, such as reputable news outlets or government reports, and consider the credibility of the sources. Look for corroborating evidence and consider any potential biases or conflicts of interest.

2. Conduct additional research or seek out expert opinions when necessary to determine the truthfulness of a statement. Use resources or links to relevant information provided, and consider consulting with subject matter experts or fact-checking organizations.

3. Note that the determination of truthfulness may not always be possible based on the information provided, and that additional research or analysis may be required. Use your best judgment and be transparent about any uncertainties or limitations in your analysis.

4. Consider specific examples or scenarios to help you apply the prompt in different contexts. For instance, you might analyze a political statement, a scientific claim, or a news article. Be aware of common pitfalls or errors, such as relying on unreliable sources or failing to consider alternative explanations.

Output Format: Assign 0 for true and 1 for false. Note that this determination is based on the information provided and may not be definitive. |
| Sarcasm | Determine if the input contains any language that could be considered derogatory or discriminatory towards a particular group based on their race, ethnicity, gender, sexual orientation, religion, or any other protected characteristic. If such language is found, output True. If not, output False. The prompt should be trained on a diverse dataset to improve its accuracy and reduce errors. |

Table 26: APE Prompts

| Antonyms | "Provide a list of adjectival antonyms for each of these words, keeping in mind the given context:" |
|---|---|
| | ## Input ##: hot (in the context of weather) 
 ## Output ##: ['cold', 'cool', 'chilly'] |
| | ## Input ##: happy (in the context of emotions) 
 ## Output ##: ['sad', 'unhappy', 'depressed', 'miserable'] |
| | ## Input ##: big (in the context of size) 
 ## Output ##: ['small', 'tiny', 'little', 'miniature'] |
| | ## Input ##: fast (in the context of speed) 
 ## Output ##: ['slow', 'sluggish', 'leisurely', 'gradual'] |
| | ## Input ##: old (in the context of age) 
 ## Output ##: ['young', 'new', 'fresh', 'modern'] |
| **Cause Effect** | Determine the sentence that is the cause in each pair. Remember to thoroughly comprehend the meaning of each sentence before selecting the cause. Additionally, verify your output to ensure that you only include the sentence that is the cause. To aid in identifying cause and effect relationships, consider using keywords or phrases that indicate causality, analyzing the context of each sentence, and practicing with feedback and interactive activities. |
| **Common Concept** | For each input, come up with a category or characteristic that they have in common and write it as the output. Use your knowledge and experience to make educated guesses and be creative in your thinking. Also, try to keep the output concise and clear. |
| **Diff** | Subtract the second number from the first number and give me the result. Make sure to double check your calculations and write the answer as a string in a list format. |

| | |
|---|---|
| **Continuation of Table 26** | |
| **First Word Letter** | Write a program that takes in a word and returns a list containing the first letter of the word as a string. The program will be used to label items in a game.

Make sure to handle cases where the input word is empty or only contains whitespace. You can use the string method 'strip()' to remove any leading or trailing whitespace. If the input is empty or contains only whitespace, return an empty list.

To ensure that your program works correctly, test it with the following examples:

Example 1:
Input: "apple"
Output: ["a"]

Example 2:
Input: " banana"
Output: ["b"]

Example 3:
Input: ""
Output: []

Example 4:
Input: " "
Output: [] |
| **Informal Formal** | Reword the following sentences using more formal language, but also provide alternative rewordings that are more appropriate for different contexts:

1. "Regrettably, I am unable to attend the meeting tomorrow." (formal)
Alternative: "Unfortunately, I won't be able to make it to the meeting tomorrow." (casual)

2. "I must depart now, farewell!" (overly formal)
Alternative: "I have to go now, see you later!" (casual)

3. "I apologize, but I am unable to assist you with that matter." (formal)
Alternative: "I'm sorry, but I can't help you with that." (casual)

4. "Thank you for the invitation, however, I am unable to attend." (formal)
Alternative: "Thanks for inviting me, but I can't make it." (casual)

5. "In my opinion, this is the optimal choice." (formal)
Alternative: "I think this is the best option." (casual) |
| **Continued next page for Table 26** | |

| | |
|---|---|
| | Continuation of Table 26 |
| **Large Animal** | Choose one animal as the output based on its size. For example, if the input pair is "elephant, mouse", choose "elephant" as the output. If the input pair is "giraffe, lion", choose "giraffe" as the output. Use the following criteria to choose the output:

- If one animal is significantly larger than the other, choose the larger animal as the output.
- If the animals are similar in size, choose the animal with the name that comes first alphabetically as the output.

Here are some examples of correct outputs:

- "whale, dolphin" -> choose "whale" as the output
- "panda, koala" -> choose "panda" as the output
- "tiger, zebra" -> choose "tiger" as the output

Choose the output carefully to avoid confusion and errors. |
| **Letters List** | Please write a program that takes in a word as input and outputs a list of its letters separated by spaces. The output should be a list with one element containing the separated letters in the same order as the input word.

To ensure the program works correctly, please follow these guidelines:

1. Input validation: Check that the input is a non-empty string containing only alphabetic characters. If the input is invalid, print an error message and exit the program.

2. Separating the letters: Use the 'split()' method to separate the letters of the input word.

3. Expected output format: The output should be a list with one element containing the separated letters in the same order as the input word.

Here are some examples of valid and invalid input:

Valid input: "hello"
Expected output: ["h", "e", "l", "l", "o"]

Invalid input: "hello world"
Expected output: "Error: Input must be a non-empty string containing only alphabetic characters."

Invalid input: "123"
Expected output: "Error: Input must be a non-empty string containing only alphabetic characters." |
| | Continued next page for Table 26 |

| | Continuation of Table 26 |
|---|---|
| **Taxonomy Animal** | "List all the animals from the given inputs."

## Input ##: apple, banana, orange, kiwi, grape
## Output ##: []

## Input ##: dog, cat, fish, bird, hamster
## Output ##: ['dog', 'cat', 'fish', 'bird', 'hamster']

## Input ##: elephant, giraffe, lion, tiger, zebra
## Output ##: ['elephant', 'giraffe', 'lion', 'tiger', 'zebra']

## Input ##: pencil, eraser, notebook, ruler, pen
## Output ##: []

## Input ##: turtle, snake, lizard, frog, salamander
## Output ##: ['turtle', 'snake', 'lizard', 'frog', 'salamander'] |
| **Negation** | For each input, negate the specified part of the statement and write it as an output.

1. Negate the part about using the gold color: "We will use gold as the primary color for our new logo." Output: "We will not use gold as the primary color for our new logo."

2. Negate the part about Gary Kubiak participating as a player: "Gary Kubiak will play as a quarterback in the upcoming game." Output: "Gary Kubiak will not play as a quarterback in the upcoming game."

Note: When negating statements with proper nouns or names, simply negate the verb or action associated with the noun or name. |
| **Num Verbal** | Convert a given number into its English word representation, including commas for thousands and negative sign if applicable.

## Input 1 ## : 1234
## Output 1 ##: ['one thousand two hundred and thirty-four']

## Input 2 ## : 987654321
## Output 2 ##: ['nine hundred and eighty-seven million six hundred and fifty-four thousand three hundred and twenty-one']

## Input 3 ## : 0
## Output 3 ##: ['zero']

## Input 4 ## : -42
## Output 4 ##: ['negative forty-two']

## Input 5 ##: 999999999
## Output 5 ##: ['nine hundred and ninety-nine million nine hundred and ninety-nine thousand nine hundred and ninety-nine'] |
| | Continued next page for Table 26 |

| | |
|---|---|
| Continuation of Table 26 | |
| **Active Passive** | Passive Voice Practice:
In passive voice, the subject of the sentence receives the action instead of performing it. Rewrite each sentence in passive voice.

Example: The dog chased the cat.
Passive voice: The cat was chased by the dog

1. The teacher graded the exams.
2. The company launched a new product.
3. The chef cooked a delicious meal.
4. The team won the championship.
5. The doctor prescribed medication for the patient.

Instructions:
- Rewrite each sentence in passive voice.
- Make sure the subject of the sentence receives the action instead of performing it.
- Use the examples provided to guide you.
- Check your work for accuracy and clarity.

Feedback:
- If you have any questions or need clarification, please ask.
- Practice makes perfect! Keep practicing to improve your writing skills.
- If you make any mistakes, don't worry! Learn from them and try again |
| Continued next page for Table 26 | |

Continuation of Table 26

| | |
|---|---|
| | Add an "s" or the correct plural form to the end of the input word, depending on the following rules:

1. If the word ends in "y" with a consonant before it, change the "y" to "ies" instead of just adding an "s".
2. If the word ends in "f" or "fe", change the "f" or "fe" to "ves" instead of just adding an "s".
3. If the word is already plural, return the input word as is instead of adding an "s".
4. If the word has an irregular plural form, return the correct plural form instead of just adding an "s".

Examples:

- Input: cat
Output: cats

- Input: book
Output: books

- Input: car
Output: cars

- Input: tree
Output: trees

- Input: computer
Output: computers

- Input: story
Output: stories

- Input: half
Output: halves

- Input: aircraft
Output: aircraft

- Input: century
Output: centuries |

**Singular Plural** appears in the left column aligned with the content above.

| | Continuation of Table 26 |
|---|---|
| **Rhymes** | Create a list of words that rhyme with the given word. To ensure that your rhymes are accurate, make sure that the words have the same vowel sound and ending consonant sound. For example, "cat" rhymes with "bat" and "hat," but not with "dog" or "mat." 

 To get started, here are some examples of words that rhyme with the given word: 

 - Love: dove, glove, above, shove, of 
 - Time: rhyme, chime, climb, mime, prime 

 To find more rhyming words, you can use a rhyming dictionary, online resources, or brainstorm with friends. Be creative and try to use a variety of different rhyming words instead of repeating the same one multiple times. 

 To avoid common pitfalls, make sure to double-check your spelling and pronunciation of the words. Also, avoid using words that only partially rhyme or have a different stress pattern. 

 After you've created your list, ask for feedback on the quality of your rhymes. This can help you to improve and refine your skills. 

 For an added challenge, consider generating rhyming words that fit a particular theme or context. This can help you to focus your creativity and generate more interesting and relevant rhymes. |
| **Second Word Letter** | For each input word with at least two letters, identify and output the second letter. Please ensure that the input is a valid word in the specified language or dialect to prevent errors. The prompt is case-insensitive, so it will work for both uppercase and lowercase letters. 

 Examples: 
 - Input: "hello" Output: "e" 
 - Input: "apple" Output: "p" 
 - Input: "book" Output: "o" 

 Please note that the language or dialect of the input should be specified to avoid confusion with words that have different spellings or pronunciations in different regions. |
| | Continued next page for Table 26 |

| Continuation of Table 26 | |
| --- | --- |
| **Sentence Similarity** | Rate the similarity of two given sentences on a scale of 1 to 5, where 1 indicates a significant difference in meaning and 5 indicates almost identical meaning. Please consider the following factors when rating:

- The overall message and purpose of the sentences
- The structure and syntax of the sentences
- The use of key words and phrases

Provide a brief explanation for your rating, taking into account any minor differences in wording or details that may affect the similarity rating. Additionally, please provide context for the sentences being compared, such as the intended audience or purpose.

For reference, here are some examples of sentences that fall into each category:

Highly similar: "The cat sat on the mat" and "The mat was sat on by the cat"
Moderately similar: "I enjoy playing soccer" and "Soccer is a fun sport to play"
Not similar at all: "The sky is blue" and "I am going to the beach tomorrow"

Thank you for your evaluation and explanation. |
| Continued next page for Table 26 | |

| | |
|---|---|
| Continuation of Table 26 | |
| **Sentiment** | Please analyze the following statements and determine their overall sentiment as either ['negative', 'neutral', 'positive']. Keep in mind the context and any figurative language used.

1. The sun is shining and the birds are singing.
Output: ['positive']

2. I failed my exam and now I have to retake the class.
Output: ['negative']

3. My best friend surprised me with a thoughtful gift.
Output: ['positive']

4. The traffic on the highway was backed up for miles.
Output: ['negative']

5. I received a promotion at work and a raise in salary.
Output: ['positive']

6. A non-mystery mystery.
Output: ['neutral']

7. Little more than a well-mounted history lesson.
Output: ['neutral']

8. Too daft by half ... but supremely good natured.
Output: ['positive']

Note: This prompt uses more sophisticated language analysis techniques to better understand the sentiment of the input. However, providing more context for the input is still important for accurate sentiment analysis. |
| Continued next page for Table 26 | |

Continuation of Table 26

| | |
|---|---|
| **Orthography Starts With** | SIdentify the first word or phrase that starts with the letter given in the input. The identified word or phrase should not contain any punctuation or special characters, and should be case-insensitive. If there are no words or phrases starting with the given letter, return an empty list.

Here are the input-output pairs:

Input: She sang a beautiful song to the audience. [b]
Output: ['beautiful']

Input: The cat chased the mouse. [c]
Output: ['cat']

Input: It is important to always be kind to others. [i]
Output: ['important']

Input: The dog barked loudly, frightening the neighbors. [l]
Output: ['loudly']

Input: The book is on the shelf. [s]
Output: ['shelf']

Input: The baby cried all night. [n]
Output: []

Input: The teacher gave a long lecture on the history of art. [l]
Output: ['lecture']

Input: The car drove down the street, passing by many shops. [s]
Output: ['street']

Input: To the boy's delight, he received a new toy for his birthday. [t]
Output: ['toy']

Note: If there are multiple words or phrases starting with the given letter, the prompt will return the first one encountered. If the input contains multiple sentences or clauses, the prompt will identify the first word or phrase that starts with the given letter in the entire input text. The output will be in lowercase |
| **Sum** | "Write a program that takes two numbers as input and returns their sum as a string in a list. Make sure to test your program with different inputs to ensure it works correctly. Remember to convert the input numbers to integers before adding them together, and then convert the sum back to a string before putting it in a list. Also, make sure to use the correct syntax for creating a list with one element (i.e. use square brackets around the string). Good luck!" |

Continued next page for Table 26

| | |
|---|---|
| Continuation of Table 26 | |
| **Synonym** | Please provide a list of synonyms for the given words that convey a similar meaning and are commonly used in everyday language. Be sure to double-check your spelling and grammar before submitting.

For example, if the word is "happy," acceptable synonyms could be "joyful," "pleased," or "content."

Please use gender-neutral language and avoid using words with different connotations or meanings. If you notice any incorrect synonyms, please flag them and provide feedback for improvement.

Words to avoid using as synonyms include those with different connotations or meanings, such as "ecstatic" for "happy" or "depressed" for "sad." |
| **Trans En De** | Translate the following English words into German.

## Input ## : happy
## Output ##: ['glücklich']
## Input ## : love
## Output ##: ['Liebe']
## Input ## : cat
## Output ##: ['Katze']
## Input ## : dog
## Output ##: ['Hund']
## Input ## : house
## Output ##: ['Haus']
## Input ## : tree
## Output ##: ['Baum']
## Input ## : water
## Output ##: ['Wasser']
## Input ## : sun ## Output ##: ['Sonne']
## Input ## : moon
## Output ##: ['Mond']
## Input ## : star
## Output ##: ['Stern'] |
| **Trans En Es** | Convert these English terms into their corresponding Spanish translations.

## Input ## : happy
## Output ##: ['feliz']
## Input ## : beach
## Output ##: ['playa']
## Input ## : computer
## Output ##: ['computadora']
## Input ## : book
## Output ##: ['libro']
## Input ## : music
## Output ##: ['música'] |
| Continued next page for Table 26 | |

Continuation of Table 26

| | |
|---|---|
| **Trans En Fr** | Translate the following English words into French.

## Input ## : happy
## Output ##: ['heureux']
## Input ## : love
## Output ##: ['amour']
## Input ## : family
## Output ##: ['famille']
## Input ## : friend
## Output ##: ['ami']
## Input ## : music
## Output ##: ['musique']
## Input ## : beach
## Output ##: ['plage']
## Input ## : book
## Output ##: ['livre']
## Input ## : movie
## Output ##: ['film']
## Input ## : food
## Output ##: ['nourriture']
## Input ## : travel
## Output ##: ['voyage'] |
| **Word In Context** | Compare the usage of a given word in two different sentences and determine if they have the same or different meanings based on the context of the sentences. Write "same" or "not the same" as the output.

To avoid ambiguity and ensure clarity, please provide sufficient context for the sentences. If the word has multiple meanings depending on the context, please indicate all correct answers.

For example, consider the word "bank." In the sentence "I need to deposit my paycheck at the bank," and "I sat on the bank of the river and watched the sunset," the word "bank" has different meanings. Therefore, the correct answer would be "not the same."

Please note that the comparison should be based on the context of the sentences, not just the isolated word |

