# OpenReview forum: "SOA: Strategic Operator Adaptation for Accelerating Joint In-Context Prompt Optimization"
_TMLR — Rejected by TMLR_

### Review · Reviewer_n29R · 2024-10-18

**Summary Of Contributions:**

This paper introduces SOA, a framework that jointly optimizes prompt instructions and few-shot examples to enhance LLM performance. SOA uses a quad-phased design to balance global exploration and local refinement. It adaptively selects effective operators and prunes weaker ones. Tested on 35 benchmarks, SOA outperforms state-of-the-art methods with 35.47% better performance and 58.67% lower computational costs.

**Audience:**

Yes

**Claims And Evidence:**

No

**Requested Changes:**

- What is **parent** mentioned in Section 3.1, there seems no explanation.
- There's no example showing the difference between SOA-pair and SOA-example, it's hard to understand these two different initialization strategies
- The overall definition of new terms is confusing,  there needs more in detail explanation for all the new terms, especially Section 3.

**Strengths And Weaknesses:**

Strengths:
1. The idea is interesting, the experiment results look good.

Weaknesses:
1. Can you explain the motivation for the quad-phase design? Why the exploration and exploitation is necessary? Which phase in most important in the quad-phase? How do they ensure that the covergence to optimal prompt?
2. Though the authors defined five operators, if I understand correctly, they are all prompting method, rather than a symbolic operator, except the semantic operator. Such statement may result in misunderstanding. Also, the authors didn't decribe how the semantic operator is implemented, how can you ensure the semantic unchanged after lexcial editing?
3. How are the five features in Section 3.1 that evalute the operator calcuated? Are they also prompt-based and relying on LLM? How did you get the results of Table 1?
4. Which operator is most effective? There's no ablation on different operators' impact on final performance.
5. There's no detail about the prompt in Table 2, are the results zero-shot or few-shot, if few-shot, how many shots? Are the baselines having the same setting? Without this information, it is hard to identify if the proposed method is effective or not.
6. How does the prompt of operator looks like? Though the whole prompt may be lengthy, but at least there should be some piece of them to give the readers an overview.

---

> ### Author Response · Authors · 2024-11-02
> **Thank you for your comments and constructive feedback.**
>
> Thank you for your comments and constructive feedback. We would like to address your questions and concerns as follows:
>
> > **W1** :Can you explain the motivation for the quad-phase design? Why the exploration and exploitation is necessary? Which phase in most important in the quad-phase? How do they ensure that the covergence to optimal prompt?
>
> Thanks for the question. The balance between exploration and exploitation has been a strategy applied in many optimization research such as Bayesian optimization. Exploration induces more cost but can ensure the border area of the problem space is traversed in search for global optimal. Exploitation can be cheaper but can be stuck at local optimal. Thus a balance between exploration and exploitation can contribute to arriving at the optimal solution efficiently.
>
> By interleaving exploration and exploitation in a quad-phase design, we aim to achieve the balance between exploration and exploitation. Thus we do not treat any of the phases particularly important. Considering that Phase 0 is initialization, we conducted an additional ablation study on all the other phases as shown in the table below.
>
> | **Task**             | **Without Phase 1** | **Without Phase 2** | **Without Phase 3** | **SOA**  |
> |----------------------|---------------------|---------------------|---------------------|----------|
> | **Disambiguation QA**| 63.71               | 66.94               | 64.52             | **72.37**|
> | **Formal Fallacies** | 50.80               | 54.03               | 52.41               | **58.87**|
>
> We observe no significant difference when removing different phases. However, removing Phase 1 with the feedback operator will cause the biggest performance degradation. We hypothesize that the feedback operator allows candidates to arrive at their local optimal efficiently. Thus removing it will cause the next phase to start with less than locally optimized candidates, which will hurt the overall performance most. Having all phases will yield the best results. This further proves the importance of orchestrating different phases.
>
>
>
> > **W2** : Though the authors defined five operators, if I understand correctly, they are all prompting method, rather than a symbolic operator, except the semantic operator. Such statement may result in misunderstanding. Also, the authors didn't decribe how the semantic operator is implemented, how can you ensure the semantic unchanged after lexcial editing?
>
> Thanks for the question. Considering the page limit we provided additional information about each operator in Appendix B. All operators are prompt-based. The individual prompt used can be found in Appendix D.
> Regarding the semantic operator, we are totally relying on LLM for the operation as other optimizers such as EvoPrompt, PromptBreeder, AELP, APE, and APO have done.

---

> ### Author Response · Authors · 2024-11-02
>
> > **W3** : How are the five features in Section 3.1 that evalute the operator calcuated? Are they also prompt-based and relying on LLM? How did you get the results of Table 1?
>
> Thanks for the question. Yes, all operators are prompt-based. To evaluate the intrinsic nature of each of them, we have conducted a series of experiments as outlined in Appendix C.2 where we ran each operator 100 times on 5 different tasks. Through these experiments, we learn information such as the likelihood of performance gain through repeated application, etc.
>
>
> > **W4** : Which operator is most effective? There's no ablation on different operators' impact on final performance.
>
> Regarding the effectiveness of each operator, we conducted additional analysis on the improvements each contributed to the experiments.
>
> | Feedback   | Avg improvement | Weighted improvement count |
> |------------|-----------------|----------------------------|
> |feedback   | 0.19            | 187                        |
> | semantic  | 0.12            | 125                        |
> | crossover        | 0.21            | 149                        |
> | eda   | 0.2             | 111                        |
>
> We observe feedback operators provide the most number of improvements, this aligns with our analysis of its ability to arrive at local optimal fast. Crossover operator, followed by EDA tend to bring the most average improvement, which aligns with our analysis of its ability to jump out of local optimal.
>
>
> > **W5** : There's no detail about the prompt in Table 2, are the results zero-shot or few-shot, if few-shot, how many shots? Are the baselines having the same setting? Without this information, it is hard to identify if the proposed method is effective or not.
>
> For the evaluation in Table 2, on the baseline approaches, we follow the exact setting as in the original paper. Methods like APE, and AELP already have a few shot examples embedded.
> For SOA, because of its ability to traverse both zero-shot and few-shot space, we have not specified how many shots the prompt should have. The full set of generated prompts can be found in Appendix H. For tasks like casual judgment, hyberbaton, etc. the final generated prompt is zero shot. For Dyke Languages, Logical Deduction Five etc. the final generated prompt is few shot. The number of few shots varies by task.
>
> We argue that for the task of automatic prompt optimization, we should evaluate optimizers based on their end-to-end capability. If an optimizer solution does not provide the ability to generate a few-shot example, we should respect the limits and compare with it as is.
> However, we have conducted an additional set of experiments adding 2 randomly selected few-shot examples from the corresponding training sets to OPRO and EvoPrompt as AELP already has few-shot examples. Below is a table reflecting the results.
>
> |                                     | SOA-pair | SOA-example | OPRO  | OPRO-fewshot | EvoPrompt | Evoprompt-fewshot | AELP  |
> |-------------------------------------|---------------|------------------|-------|--------------|-----------|-------------------|-------|
> | disambiguation_qa                   | 72.13         | 68.47            | 71.53 | 66.93        | 53.7      | 57.43             | 62.69 |
> | formal_fallacies                    | 58.87         | 58.65            | 49.51 | 52.41        | 50.81     | 43.54             | 57.95 |
> | hyperbaton                          | 86.02         | 87.5             | 75.92 | 62.9         | 74.79     | 79.83             | 52.64 |
> | salient_translation_error_detection | 49.19         | 47.59            | 43.88 | 37.39        | 47.58     | 31.45             | 38.93 |
>
> With randomly added few-shot examples, OPRO observes performance gain in 1 / 4 tasks and EvoPrompt 2 / 4. This test further confirms that by naively adding randomly selected few-shot examples, there is a risk for performance degradation. Thus a unified method of joint optimization of instruction and examples like SAO does is critical.

---

> ### Author Response · Authors · 2024-11-02
>
> > **W6** : How does the prompt of operator looks like? Though the whole prompt may be lengthy, but at least there should be some piece of them to give the readers an overview.
>
> The full set of operator prompts can be found in Appendix D.
>
> > **C1** : What is parent mentioned in Section 3.1, there seems no explanation.
>
> Thanks for the question. For each stage, we are essentially applying operators to generate new candidates. Considering the candidates are evolved based on previous ones, we treat the previous prompt candidate that is involved in generating the new candidate as the new candidate’s parents. We will clarify that in the final version.
>
> > **C2** :There's no example showing the difference between SOA-pair and SOA-example, it's hard to understand these two different initialization strategies
>
> Thanks for the insight. Both SOA-pair and SOA-example aim to optimize the prompt instruction and few-shot tunings jointly. We design SOA with the intention of making it widely applicable and useful across various use cases. And one thing we note in real applications is that not all cases will have the same starting point. Some use cases may only have access to input-output pairs for the task, while others may only have human experts for human-composed prompts. In fact, similar patterns have been observed in other studies, such as AELP, which relies on human-composed prompts, and OPRO, which only uses input-output pairs. To address this variability, we have designed two initialization methods for SOA. SOA-pair employs input-output pairs of the task as its initialization method, while SOA-example takes human-composed example prompts as the starting point. We will supplement the information with additional clarity beyond what we had in the final version.
>
> > **C3** :The overall definition of new terms is confusing, there needs more in detail explanation for all the new terms, especially Section 3.
>
> Thanks for the question. We agree having new terms is confusing and have provided more details for the definitions in Appendix B.
>
> We believe that our responses address your concerns and we hope that you will reconsider the paper. We are dedicated to making improvements to the paper based on your input. Please let us know if you need further clarification or any other comments!

---

### Review · Reviewer_Qcmc · 2024-10-21

**Summary Of Contributions:**

This paper introduces SOA (Strategic Operator Adaptation), a novel framework for optimizing prompts for large language models (LLMs) that jointly tunes both instructions and few-shot examples. The authors present a formulation for joint optimization of instructions and examples, rather than treating them separately as in prior work that alternates between global exploration and local exploitation.

The authors introduce multiple operators to traverse the prompt space effectively by adaptive selection of operators and pruning of candidates to accelerate convergence.

The main results demonstrate that SOA outperforms baselines like APE, APO, OPRO, PromptBreeder, EvoPrompt and AELP by large margins, especially on harder tasks. For example, on BIG-Bench-Hard tasks, SOA achieves a 35.47\% average improvement in accuracy while reducing inference costs by 58.67\%. However, this framework has heavy reliance on LLMs as operators (which is a significant limitation given it boils down to LLM suggesting something to LLM). There is a lack of clarity on computational costs and inference details and very few studies on individual components.

**Audience:**

Yes

**Claims And Evidence:**

Yes

**Requested Changes:**

1. Provide full implementation details for all LLM-based operators, including exact prompts used. This is crucial for reproducibility.
2. Explore and report on the sensitivity of SOA to key hyperparameters and initialization strategies.
3. Include more information on the operators and provide ablation studies on each of them.

**Strengths And Weaknesses:**

## Strengths

1. The adaptive selection of operators based on their performance allows choosing the most effective strategies at different optimization stages, contributing to efficiency and effectiveness.
2. The extensive evaluation on 35 benchmark tasks demonstrates significant improvements over state-of-the-art baselines, especially on harder tasks.
3. The authors provide useful analysis of operator behaviors, prompt length variations, and synthetic example generation, giving insight into how SOA works.
4. SOA can work with both zero-shot and few-shot prompts, and can add or remove examples dynamically. This flexibility is valuable for adapting to different tasks.

## Weaknesses

1. The paper claims significant reductions in computational costs but does not provide a clear breakdown or analysis of the total costs involved, including the LLM calls for operators.
2. No information is provided on how the feedback gradients are calculated for Phase 1. LLM examination and improvement could potentially be done by using just one agent rather than doing two tasks, which may not reduce the overall computation as the authors claim.
3. The method involves many design choices and hyperparameters (e.g., pool sizes, operator tolerances). The sensitivity to these choices is not thoroughly explored.
4. It is unclear how SOA would interact with Chain of Thought (CoT) approaches, which have been successful in deducing correct (or almost correct) prompts from users in recent papers.

---

> ### Author Response · Authors · 2024-11-02
> **Thank you for your comments and constructive feedback**
>
> Thank you for your comments and constructive feedback. We would like to address your questions and concerns as follows:
>
> > **W1** : The paper claims significant reductions in computational costs but does not provide a clear breakdown or analysis of the total costs involved, including the LLM calls for operators.
>
> Thanks for the question, We have analyzed the computational cost in Figure 6. Computational cost is calculated using two metrics: the total iteration and the total number of calls to LLM during the optimization process. The total number of calls to LLM includes both the calls made to apply the operators, as well as the calls made for evaluations of the candidates.
>
> We intentionally did not separate between LLM calls for operators and evaluations for the comparison as we consider it reasonable to evaluate the end-to-end cost. If a method needs significantly more evaluation data for the end-to-end process, even though it does not cost as much in operators, the total cost is still high to achieve similar performance and vice versa.
>
> SOA is able to finish optimization efficiently, evidenced by the y-axis in Figure 6, which shows an average of 12.32 iterations for an end-to-end process. On the other hand, approaches like AELP require 50 iterations, while OPRO takes 150-200 iterations.
> The x-axis in Figure 6 indicates that SOA(star) makes significantly fewer API calls to LLM for end-to-end optimization, resulting in lower overall costs than other optimizers. For instance, the total API cost for end-to-end SOA is around 4000, whereas AELP adds up to around 10,000 and OPRO can exceed 100,000.
>
> In summary, SOA improves efficiency with respect to both the time required to complete iterations and the cost associated with API calls. We also list the breakdown of calls for operators vs calls for evaluations in the table below. We can add this table to the final version and make the cost analysis more clear.
>
>
> |                | APO   | APE   | PromptBreeder | EvoPrompt | OPRO  | AELP  | SOA   |
> |----------------|-------|-------|---------------|-----------|-------|-------|-------|
> | **Operator LLM call percentage** | 7.7%  | 0.1%  | 2%            | 7.4%      | 1.6%  | 2%    | 1.4%  |
> | **Evaluation LLM call percentage** | 92.3% | 99.9% | 98%           | 92.6%     | 98.4% | 98%   | 98.6% |
>
>
> > **W2** : No information is provided on how the feedback gradients are calculated for Phase 1. LLM examination and improvement could potentially be done by using just one agent rather than doing two tasks, which may not reduce the overall computation as the authors claim.
>
> We have provided the full prompt of the feedback operator in Table 12 and Table 13. Regarding utizliation two agents for the task we are following the setup for APO[1] for the operator setup to achieve a fair comparison. We agree having one agent can potentially do the same task and consider it a great topic for the future.
>
> For overall computation we evaluate it by the total number of API cost consumed in the optimization process, instead of focusing on the API cost of one operator. We have demonstrated the efficiency of SOA over other optimizers in Figure 6. Compared to the other baseline, SOA is the most cost-effective method that significantly reduces multiple orders of magnitude compared to evolution strategies. For example, PromptBreeder is approximately 2.5 orders of magnitude in terms of LLM calls compared to SOA.
>
> [1] Pryzant, et al. 2023, Automatic Prompt Optimization with “Gradient Descent” and Beam Search

---

> ### Author Response · Authors · 2024-11-02
>
> > **W3** : The method involves many design choices and hyperparameters (e.g., pool sizes, operator tolerances). The sensitivity to these choices is not thoroughly explored.
>
> For the hyperparameters like evaluation data size, we have been following what the previous researchers have done and set it to 50.
>
> The predefined threshold for phase transition and the pool size are parameters that can be configured. Regarding the universal applicability of these parameters, we have utilized a threshold of 1% and a pool size of 15 for initialization, and 5 for the rest of the phases in all 35 tasks and achieved superior results without specifically calibrating it. Similarly, in the additional experiments conducted on 7 other models such as GPT-4, claude2, palm2, llama2-7b, llama3-8b, llama3-70b, and mistral-7b, this same configuration provided competitive results. In our opinion, little or no effort is required to tune these parameters for SOA.
>
> In real applications, parameters like evaluation data size and pool size depend on the use case. If there is more data for evaluation, it will enhance the evaluation and thus lead to better performance. Similarly were there more candidates in the pool, just like evolution algorithms, it would lead to better performance. Through experimentation, we have demonstrated that even with a modest evaluation data size of 50 and a pool size of 5, SOA is able to achieve better than SOTA performance. Should the parameters increase, the overall performance of SOA would be better.
>
>
> > **W4** : It is unclear how SOA would interact with Chain of Thought (CoT) approaches, which have been successful in deducing correct (or almost correct) prompts from users in recent papers.
>
> The goal of SOA is to handle prompt optimization automatically from end to end, thus we are not focusing on any specific techniques, whether it is CoT or few-shot examples. We agree CoT is successful in a lot of tasks, and there are also research on not applying CoT to all cases universally as it can hurt performance. [1][2]
>
> In Table 24 we have listed the generated prompts where we have observed:
>
> For the task Logical Deduction Five, there is an explanation section in the prompt that adopts CoT.
>
> For the task Reasoning Colored Objects, the technique of “Let’s think step by step” is generated.
>
> For the task Salient Tranlation Error Detection, a few shot examples are generated.
>
> In summary, SOA has proven to have the ability to apply CoT and other techniques, provided that they do help with the overall performance.
>
> [1] Sprague, et al. 2023, To CoT or not to CoT? Chain-of-thought helps mainly on math and symbolic reasoning
>
> [2] Wei, et al. 2022, Chain-of-Thought Prompting Elicits Reasoning in Large Language Models
>
>
> > **C1** : Provide full implementation details for all LLM-based operators, including exact prompts used. This is crucial for reproducibility.
>
>
> We agree it is crucial to provide full details of LLM-based operators, including exact prompts. We have provided them in Appendix D.
>
> > **C2** :Explore and report on the sensitivity of SOA to key hyperparameters and initialization strategies.
>
> We conducted SOA with  35 tasks and achieved superior results without specifically calibrating it. Similarly, in the additional experiments conducted on 7 other models such as GPT-4, claude2, palm2, llama2-7b, llama3-8b, llama3-70b, and mistral-7b, this same configuration provided competitive results. In our opinion, little or no effort is required to tune these parameters for SOA.
> Regarding initialization strategies, we have conducted experimentation in all BBH tasks as illustrated in Table 2. Both strategies have tasks where they are better suited and there is no clear conclusion about what strategy is universally better.

---

> ### Author Response · Authors · 2024-11-02
>
> > **C3** :Include more information on the operators and provide ablation studies on each of them.
>
> More information on the operators can be found in Appendix B section where we listed the definition and usage for each operator.
> Regarding the effectiveness of each operator, we add additional analysis on the improvements each contributed in all the experiments.
>
> | Feedback   | Avg improvement | Weighted improvement count |
> |------------|-----------------|----------------------------|
> |feedback   | 0.19            | 187                        |
> | semantic  | 0.12            | 125                        |
> | crossover        | 0.21            | 149                        |
> | eda   | 0.2             | 111                        |
>
> We observe feedback operators provide the most number of improvements, this aligns with our analysis of its ability to arrive at local optimal fast. Crossover operator, followed by EDA tend to bring the most average improvement, which aligns with our analysis of their ability to jump out of local optimal.
>
> As operators are linked to phases, we also conducted additional ablation studies on each phase.
> | **Task**             | **Without Phase 1** | **Without Phase 2** | **Without Phase 3** | **SOA**  |
> |----------------------|---------------------|---------------------|---------------------|----------|
> | **Disambiguation QA**| 63.71               | 66.94               | 64.52             | **72.37**|
> | **Formal Fallacies** | 50.80               | 54.03               | 52.41               | **58.87**|
>
> We observe no significant difference when removing different phases. However, removing Phase 1 with the feedback operator will cause the biggest performance degradation. We hypothesize that the feedback operator allows candidates to arrive at their local optimal efficiently. Thus removing it will cause the next phase to start with less than locally optimized candidates, which will hurt the overall performance most. Having all phases will yield the best results. This further proves the importance of orchestrating different phases. We will add this to the final version.
>
> We believe that our responses address your concerns and we hope that you will reconsider the paper. We are dedicated to making improvements to the paper based on your input. Please let us know if you need further clarification or any other comments!

---

### Review · Reviewer_9xBi · 2024-10-29

**Summary Of Contributions:**

This paper presents the Strategic Operator Adaptation (SOA) framework for optimizing prompt-tuning in Large Language Models (LLMs), addressing prompt instructions and in-context examples simultaneously. Traditional prompt-tuning approaches typically separate these elements, resulting in suboptimal performance and requiring significant computational resources and expertise. SOA improves computational efficiency and optimization effectiveness in this high-dimensional prompt-tuning space. The framework uses a four-phase strategy that alternates between global exploration and local optimization, enabling efficient navigation of the complex prompt space. SOA also dynamically selects effective operators and prunes ineffective ones, speeding up convergence and reducing inference costs. Extensive experiments on various benchmarks demonstrate the efficacy of SOA, showing that it outperforms many existing baseline methods.

**Audience:**

Yes

**Broader Impact Concerns:**

There is no Broader Impact concern found in the current draft.

**Claims And Evidence:**

Yes

**Requested Changes:**

As discussed in the “Weaknesses” section above:

- It is important to provide additional justification for why Algorithm 1 solves the optimization problem in Equation (1).
- It would be a significant improvement to include theoretical guarantees for the proposed algorithm in SOA.
- To improve understanding of the proposed framework, it would be beneficial to include a concrete running example of SOA, such as detailing the starting prompt, intermediate prompts produced by SOA, and the final prompt. Essentially, this would expand upon the example in Figure 1.
- It is essential to report the computational overhead introduced by SOA across the various LLMs used in the experiments.

**Strengths And Weaknesses:**

Strengths:
- The paper is well-written and well-motivated.
- Formulating prompt-tuning of LLMs as an optimization problem and developing automated LLM prompt-tuning methods are promising research directions.
- The experimental results of the proposed SOA framework appear promising.

Weaknesses:
- There seems to be a gap between the proposed algorithm and the formulated optimization problem. It is not clear why the algorithm necessarily maximizes the objective function in Equation (1).
- There is no theoretical guarantee that the algorithm optimizes the formulated objective function.
- The computational overhead (measured in terms of runtime) of the proposed SOA framework is not clearly explained.

---

> ### Author Response · Authors · 2024-11-02
> **Thank you for your comments and constructive feedback**
>
> Thank you for your comments and constructive feedback. We would like to address your questions and concerns as follows:
>
> > **W1** : There seems to be a gap between the proposed algorithm and the formulated optimization problem. It is not clear why the algorithm necessarily maximizes the objective function in Equation (1).
>
> Thanks for the question. The objective function in Equation(1) is essentially optimizing the performance of LLM prompt based on a task. Similarly to how evolution algorithms aim to optimize an objective function around the fitness score of candidates, we consider the performance of LLM on the task to be the fitness score, and the generated prompts to be candidates in evolution algorithms. The effect of the SOA algorithm on the objective function, or “fitness score” is further testified in Figure 3.
>
> Unlike traditional evolutionary algorithms used by baselines such as EvoPrompt and PromptBreeder, SOA employs a phased design that interleaves global and local optimization to achieve the balance between exploration and exploitation, which has been essential to optimization problems in various research. This approach allows SOA to optimize efficiently with clear targets in each phase, rather than relying solely on random mutations for optimization in traditional evolutionary algorithms. This strategic change significantly reduces LLM calls, as demonstrated in Figure 6.
>
>
>
> > **W2** :There is no theoretical guarantee that the algorithm optimizes the formulated objective function.
>
> We agree that theoretical guarantee can be helpful and for algorithms like evolutionary algorithms, it is challenging to analyze in the same way as deterministic algorithms because of the degree of randomness. The same condition applies to SOA due to the usage of LLMs as operators. And the theoretical convergence in EA requires an infinite population and is not practical in real-world scenarios.
>
> Beyond evolutionary algorithms, existing literature on prompt optimization encounters the same issue due to the randomness of LLM and practical considerations.[1][2][3][4]. We observe in the border research area of LLM previous works, such as Chain-of-Thought Prompting Elicits Reasoning in Large Language Models [5] and Large Language Models are Zero-Shot Reasoners [6], employ critical empirical approaches to help understand and harness the power of LLMs other than relying on theoretical proof.
>
> [1] Zhou Yongchao et al. 2023, Large Language Models Are Human-Level Prompt Engineers
>
> [2] Fernando Chrisantha et al. 2023, Promptbreeder: Self-Referential Self-Improvement Via Prompt Evolution
>
> [3] Yuan Chengrun et al. 2024, Large Language Models as Optimizers
>
> [4] Qingyan Guo et al. 2023, Connecting Large Language Models with Evolutionary Algorithms Yields Powerful Prompt Optimizers
>
> [5] Wei et al., 2023. Chain-of-Thought Prompting Elicits Reasoning in Large Langu

---

> ### Author Response · Authors · 2024-11-02
>
> > **W3** :The computational overhead (measured in terms of runtime) of the proposed SOA framework is not clearly explained.
>
> Thanks for the question, We analyze the computational cost in Figure 6. Computational cost is calculated using two metrics: the total iteration and the total number of calls to LLM during the optimization process. The total number of calls to LLM includes both the calls made to apply the operators, as well as the calls made for evaluations of the candidates. We intentionally choose the total number of calls to LLM as it is correlated to the time it takes to run the optimization process from end to end.
>
> SOA is able to finish optimization efficiently, evidenced by the y-axis in Figure 6, which shows an average of 12.32 iterations for an end-to-end process. On the other hand, approaches like AELP require 50 iterations, while OPRO takes 150-200 iterations.
> The x-axis in Figure 6 indicates that SOA(star) makes significantly fewer API calls to LLM for end-to-end optimization, resulting in lower overall costs than other optimizers. For instance, the total API cost for end-to-end SOA is around 4000, whereas AELP adds up to around 10,000 and OPRO can exceed 100,000.
>
> In summary, SOA improves efficiency with respect to both the time required to complete iterations and the cost associated with API calls. We also list the breakdown of calls for operators vs calls for evaluations in the table below. We can add this table to the final version and make the cost analysis more clear.
>
> |                | APO   | APE   | PromptBreeder | EvoPrompt | OPRO  | AELP  | SOA   |
> |----------------|-------|-------|---------------|-----------|-------|-------|-------|
> | **Operator LLM call percentage** | 7.7%  | 0.1%  | 2%            | 7.4%      | 1.6%  | 2%    | 1.4%  |
> | **Evaluation LLM call percentage** | 92.3% | 99.9% | 98%           | 92.6%     | 98.4% | 98%   | 98.6% |
>
>
> > **C1** : It is important to provide additional justification for why Algorithm 1 solves the optimization problem in Equation (1).
>
> We consider SOA to follow the general principle of the evolutionary algorithms in terms of iteratively updating the candidate pool and evaluating candidates on fitness scores. The fitness score is essentially what Equation (1) represents.
>
> Unlike traditional evolutionary algorithms used by baselines such as EvoPrompt and PromptBreeder, SOA employs a phased design that interleaves global and local optimization and applies operators accordingly to their intrinsic advantages. This design allows SOA to achieve better performance with shorter convergence times as illustrated in Table 2 and Figure 6.
>
> Regarding the effectiveness of Algorithm1 for the problem in Equation 1, we demonstrated the score changes through the process in Figure 3, where SOA demonstrates significant and steady improvement as an empirical justification of SOA.
>
> > **C2** :It would be a significant improvement to include theoretical guarantees for the proposed algorithm in SOA.
>
> Thanks for the comment. We agree that theoretical analysis has value and that LLM optimization is challenging due to the current state of knowledge on how LLMs operate. For non-deterministic algorithms, whether they use LLM or not, theoretical analysis can be challenging and not practical in the real world, such as for evolutionary algorithms.
>
> > **C3** :To improve understanding of the proposed framework, it would be beneficial to include a concrete running example of SOA, such as detailing the starting prompt, intermediate prompts produced by SOA, and the final prompt. Essentially, this would expand upon the example in Figure 1.
>
> We agree that examples can be of great value. Considering that the process can generate over 50 prompts, we have listed the final generated prompts for tasks in Appendix H. For intermediate prompt and to show the before and after effect of operators, we listed specific examples in Appendix C-1, and Appendix G. specifically about few-shot examples.
>
> > **C4** :It is essential to report the computational overhead introduced by SOA across the various LLMs used in the experiments.
>
> We analyzed the computational cost in Figure 6. Computational cost is calculated by two metrics: the total iteration and the total number of calls to LLM during the optimization process. The total number of calls to LLM includes both the calls made to apply the operators, as well as the calls made for evaluations of the candidates.  Compared to the other baseline, SOA is the most cost-effective method that significantly reduces multiple orders of magnitude compared to evolution strategies. For example, PromptBreeder is approximately 2.5 orders of magnitude in terms of LLM calls compared to SOA.
>
>
> We believe that our responses address your concerns and we hope that you will reconsider the paper. We are dedicated to making improvements to the paper based on your input. Please let us know if you need further clarification or any other comments

---

### Decision · Action_Editor_B9DX · 2024-12-09

**Recommendation:** Reject

**Comment:**

All three reviewers agree that the proposed idea is novel and interesting, and that the experimental results look good. However, they also highlighted some weaknesses regarding reproducibility, theoretical guarantees, clarity of the computational costs, and the lack of an ablation study. Following discussions with the authors some of these weaknesses seem to have been addressed, but some also seem to remain and the reviewers did not reach consensus in their final recommendations. After a careful examination of the initial reviews, the authors' answers, and a detailed inspection of the paper, I am unfortunately leaning towards a rejection. In its current version the manuscript still has significant clarity and experimental issues which must be addressed to convincingly support the paper's claims, and satisfy the acceptance criteria for TMLR. I strongly encourage the authors revise the paper accordingly, and re-submit as I do believe the proposed method is of interest and could be valuable to the community.

On top of the various comments made by the reviewers, below are my own comments which I think the authors should address:

The current version of the paper has several presentation issues:
 - Section 3.1 introduces new terms without a definition, which are left for the reader to figure out. What is meant by candidates (pairs? prompts?), parents?, distinctiveness?, pairs of tasks?
 - Section 3.2 is unclear and key details about each operator are missing. From the text it is not clear which operators require a full performance evaluation of the prompts, and which part of the dataset is used for that (training / validation). It is also unclear how operators distinguish instructions from few-shot examples in the prompts (the concepts presented in eq. 1).
 - Algorithm 1 contains typos and ambiguities: what is Ddev versus Dtrain and Dtest? What are t and t* ? How are they updated? What is k? These notations are inconsistent also with respect to training / evaluation / testing in Appendix E2.
 - Figures are unclear: what are the score values in Fig. 3? Are they computed on the training / validation / test set? Which exact method / LLM is reported in Fig. 3? In Figure 6, which datasets are these statistics computed for?
 - Inconsistencies: In Section 4.2 the text says SOA uses GPT-3.5, but the tables mention GPT-4 as well. Two methods SOA-pair and SOA-example appear in 4.2 and 4.3 before they are described. Their description in 4.3 is also vague about the implementation of each method.

The reported experimental results do not cover the claimed 35 benchmarks, but only specific subsets: 3 benchmarks in Figure 3; 8 tasks in Table 2 vs OPRO, EvoPrompt, AELP; 3 tasks in Table 3 vs APO; 8 tasks in Figure 4 vs APE and PromptBreeder; 4 tasks in Table 4; 4 tasks (different subset) in Table 5; 24 tasks in Table 18 (appendix). In Figures 5 and 6, the task subset is not even reported. Why are all methods / variants not evaluated on the entire set of 35 tasks?

Finally, an ablation study that measures the effect of the proposed adaptive phase stop criteria is missing.

**Audience:**

All reviewers agree that the topic studied in this work is of interest to the community.

**Claims And Evidence:**

This paper entitled "SOA: Strategic Operator Adaptation for Accelerating Joint In-Context Prompt Optimization" introduces a new method for prompt optimization, which is described and evaluated on a series of question-answering benchmarks. The claims are the following:
 - the proposed method outperforms state-of-the-art baseline methods by a large margin, while reducing the computational costs
 - the improvements are imputable to
   - the proposed quad-phased design, which strikes an optimal balance between exploration and exploitation
   - the proposed performance-vector similarity metric, which is more effective than traditional lexical similarity metrics
   - the proposed adaptive phase stop criteria, which ensures maximum performance improvement within each operator

After careful examination of the reviews, the authors' responses and the paper, I believe that paper does not provide convincing and clear evidence for its claim. See the comments section for more details.

**Resubmission Of Major Revision:**

The authors may consider submitting a major revision at a later time.